# Minimax Optimal Algorithms for Fixed-Budget Best Arm Identification

**Junpei Komiyama**
New York University
New York, NY, United States
junpei@komiyama.info

**Taira Tsuchiya**
Kyoto University, Kyoto, Japan
RIKEN AIP, Tokyo, Japan
tsuchiya@sys.i.kyoto-u.ac.jp

**Junya Honda**
Kyoto University, Kyoto, Japan
RIKEN AIP, Tokyo, Japan
honda@i.kyoto-u.ac.jp

## Abstract

We consider the fixed-budget best arm identification problem where the goal is to find the arm of the largest mean with a fixed number of samples. It is known that the probability of misidentifying the best arm is exponentially small to the number of rounds. However, limited characterizations have been discussed on the rate (exponent) of this value. In this paper, we characterize the minimax optimal rate as a result of an optimization over all possible parameters. We introduce two rates, $R^{\mathrm{go}}$ and $R^{\mathrm{go}}_\infty$, corresponding to lower bounds on the probability of misidentification, each of which is associated with a proposed algorithm. The rate $R^{\mathrm{go}}$ is associated with $R^{\mathrm{go}}$-tracking, which can be efficiently implemented by a neural network and is shown to outperform existing algorithms. However, this rate requires a nontrivial condition to be achievable. To address this issue, we introduce the second rate $R^{\mathrm{go}}_\infty$. We show that this rate is indeed achievable by introducing a conceptual algorithm called delayed optimal tracking (DOT).

## 1 Introduction

We consider $K$-armed best arm identification problem with $T$ samples. In this problem, each arm $i \in [K] = \{1, 2, \ldots, K\}$ is associated with (unknown) distribution $P_i \in \mathcal{P}$ for some class of distributions $\mathcal{P}$. Upon choosing arm $i$, the forecaster observes reward $X(t)$, which is drawn independently from $P_i$. The forecaster then tries to identify (one of) the best arm[1] $\mathcal{I}^* = \arg\max_i \mu_i$ with the largest mean $\mu^* = \max_i \mu_i$ for $\mu_i = \mathbb{E}_{X \sim P_i}[X]$. The problem[2] is called the best arm identification (BAI, Audibert et al. (2010)), or the ranking and selection (R&S, Hong et al. (2021)).

To this aim, the forecaster uses some algorithm that would adaptively choose an arm based on its history of rewards. At each round $t$, the algorithm chooses one of the arms $I(t) \in [K]$ and receives the corresponding reward $X(t)$. After the $T$-th round, the algorithm outputs a recommendation arm $J(T) \in [K]$, which corresponds to an estimator of the best arm. The probability of misidentification is expressed by $\mathbb{P}[J(T) \notin \mathcal{I}^*]$, which will be referred to as

---

[1]We use $\mathcal{I}^* = \mathcal{I}^*(\boldsymbol{P}) \subset [K]$ as the set of best arms and $i^* = i^*(\boldsymbol{P}) \in \mathcal{I}^*(\boldsymbol{P})$ as one of them (ties are broken in an arbitrary way). These differences do not matter much in this paper.

[2]See Section 1.3 regarding the related work on BAI and R&S.

36th Conference on Neural Information Processing Systems (NeurIPS 2022).

the probability of the error (PoE) throughout the paper. Best arm identification has two settings. In the fixed confidence (FC) setting, the forecaster minimizes the number of draws $T$ until the confidence level on the PoE reaches a given value $\delta \in (0, 1)$. In this case, $T$ is a stopping time that can be chosen adaptively. In the fixed-budget (FB) setting, the forecaster tries to minimize the PoE given a constant $T$. In this paper, we shall focus on the FB setting. In general, a good algorithm for the FC setting is very different from that for the FB setting. To be more specific, an algorithm for the FC setting can be uniformly optimal.[3] Namely, several FC algorithms exist (Garivier and Kaufmann, 2016) that is able to adapt to each instance of distributions $\boldsymbol{P} = (P_1, P_2, \ldots, P_K)$. On the contrary, an algorithm for the FB setting requires consideration of the tradeoff in that improving the PoE for an instance $\boldsymbol{P}$ can compromise the PoE for another instance $\boldsymbol{P}'$. Thus, we must consider an optimization of the performance over all possible $\boldsymbol{P} \in \mathcal{P}^K$. See Appendix C for a demonstration of the asymptotic inconsistency between the two settings.

## 1.1 Minimax optimality in the fixed-budget setting

In the FB setting, the PoE decays exponentially to $T$ as $\exp(-RT)$ for some *rate $R > 0$*. The uniform optimality given above is no longer available here. To demonstrate this, assume that we make an estimate of $\boldsymbol{P}$ based on the initial $o(T)$ rounds, say, $\sqrt{T}$ rounds. In this case, we can obtain the estimate of $\boldsymbol{P}$ that is $\epsilon$-correct with probability $\exp(-\epsilon^2 O(\sqrt{T})) = \exp(-o(T))$. However, this estimation does not help to improve the rate of exponential convergence. In other words, estimating $\boldsymbol{P}$ requires non-negligible (i.e. $O(T)$) cost for exploration. As a result, we cannot fully adapt the PoE to each instance $\boldsymbol{P}$ unlike the FC setting. Instead, to discuss optimality in the FB setting, we must choose a complexity function $H(\boldsymbol{P})$, and the performance of an algorithm must be evaluated on the rate normalized by the complexity $\exp(-RT/H(\boldsymbol{P}))$.

In the literature, little is known about the optimal rate of the exponent. Several papers considered what rate $R$ is achievable for all $\boldsymbol{P}$ given a complexity function $H(\boldsymbol{P})$. Kaufmann et al. (2016) showed that this rate is larger than the corresponding rate of the FC setting.[4] Audibert et al. (2010) proposed the successive rejects (SR) algorithm, which has the rate of $1/(\log K)$ with the complexity function $H_2(\boldsymbol{P}) := \max_{i \in [K]} \frac{i}{\Delta_i^2}$ for $\Delta_i = \max_j \mu_j - \mu_i$ satisfying $\Delta_1 \leq \Delta_2 \leq \cdots \leq \Delta_K$. Carpentier and Locatelli (2016) showed a particular set of instances such that this rate matches the lower bound up to a constant factor. However, the constant used there is by far loose[5], and there is limited discussion on the actual rate of such algorithms.

## 1.2 Contributions

This paper tightly characterizes the optimal minimax rate of the PoE as a result of an optimization given $\boldsymbol{P}$. Let $H = H(\boldsymbol{P}) : \mathcal{P}^K \to \mathbb{R}^+ \cup \{\infty\}$ be any complexity measure. We then discuss the best possible rate $R > 0$ such that the PoE is bounded by $\exp(-RT/H(\boldsymbol{P}) + o(T))$ for all $\boldsymbol{P} \in \mathcal{P}^K$ and make the following contributions.

- We derive an upper bound on $R$ (corresponding to a lower bound of the PoE), denoted by $R^{\mathrm{go}}$, which we obtain by considering a class of oracle algorithms that can determine the allocation of trials to each arm knowing the final empirical distribution after $T$ rounds (Theorem 1).

- We propose an algorithm ($R^{\mathrm{go}}$-tracking) that tracks this oracle allocation based on the current empirical distribution (Section 2.1). Although this oracle allocation is expressed by a complicated minimax optimization, we propose a technique to learn this by a neural network and empirically confirm that the PoE of the learned algorithm is close to the lower bound (Sections 3 and 4). We also discuss that the algorithm is

---

[3] A more complete discussion on this topic can be found in Section B.

[4] That is, given the complexity of the FC setting, the rate for some instances is smaller than 1 in the FB setting.

[5] Theorem 1 therein includes a large constant 400.

unlikely to provably achieve the bound even when the minimax problem is perfectly solved because of the impossibility of tracking.

- We tighten the PoE lower bound by weakening the oracle algorithms to obtain a new rate $R_\infty^{\mathrm{go}}$. We propose the delayed optimal tracking (DOT) algorithm that asymptotically achieves this rate for Bernoulli and Gaussian arms. While DOT is minimax, the algorithm is computationally almost infeasible (Sections 2.2 and 2.3).

In summary, we propose a nearly tight minimax lower bound rate on the PoE with a computationally feasible algorithm that is empirically close to this bound. We also propose exact minimax rate and a matching algorithm in a computationally infeasible form. Notation is listed in the Section A in the appendix.

## 1.3 Related work

Compared with the works of the fixed-confidence (FC) BAI, less is known about the fixed-budget (FB) BAI. For example, a book on this subject (Lattimore and Szepesvári, 2020) spends only two pages on the FB-BAI.[6] Many algorithms designed for the FC-BAI, such as D-tracking (Kaufmann et al., 2016), do not have a finite-time PoE guarantee when we apply them to the FB setting. Nevertheless, there are two well-known FB BAI algorithms: Successive rejection (SR, Audibert et al. (2010)) and successive halving (SH, Shahrampour et al. (2017)). Both SR and SH decompose $T$ time steps to a finite number of time segments, and then progressively narrow the candidate of the best arm at the end of each segment. While SR discards one arm after each segment, SH discards half of the remaining arms after each segment. SR and SH have the guarantee on the PoE of the rate $\exp\left(-RT/H_2(\boldsymbol{P})\right)$ for some constant $R > 0$. Other FB BAI algorithms, such as UCB-E (Audibert et al., 2010) and UGapE (Gabillon et al., 2012), require the knowledge of minimum gap $\min_i \Delta_i$, and thus are not universal to all best arm identification instances.

Another literature on this topic is the ranking and selection (R&S) problems (Bechhofer and Sobel, 1954; McDonald, 1998; Powell and Ryzhov, 2018; Hong et al., 2021). Although the goal of R&S problems is to identify the best arm, many R&S papers do not consider the estimation error of $\boldsymbol{P}$ in a finite time. As a result, algorithms therein do not have the guarantee on the PoE in the best arm identification setting. The optimal computing budget allocation (OCBA, Chen et al. (2000); Glynn and Juneja (2004)) algorithm tries to minimize the PoE assuming the plug-in estimator matches the true parameter. Bayesian R&S algorithms try to solve the dynamic programming of minimizing the PoE marginalized by a prior (Bayesian PoE), which is computationally prohibitive, and thus approximated solutions have been sought (Frazier et al., 2008; Powell and Ryzhov, 2018).

## 2 Minimax optimal algorithm

In this section, we derive several lower bounds on the PoE and propose algorithms to empirically or theoretically achieve these bounds.

First, we formalize the problem. Let $\mathcal{P}$ be a known class of reward distributions. We consider the case where $\mathcal{P}$ is the set of Bernoulli distributions with mean $\Theta \subset [0,1]$ (including the case $\Theta = [0,1]$), or Gaussian distributions with mean in $\Theta \subset \mathbb{R}$ (including the case $\Theta = \mathbb{R}$) and known variance $\sigma^2 > 0$. It should be noted that many parts of the results in this paper can be generalized to much wider classes of distributions, but it makes the notation much longer and is discussed in Appendix D.

When we derive lower bounds and construct algorithms, we introduce $\mathcal{Q}$ as a class of distributions corresponding to the estimated distributions of the arms. Namely, we set $\mathcal{Q}$ as the set of all Bernoulli (resp. Gaussian) distributions with mean in $[0,1]$ (resp. $\mathbb{R}$) when $\mathcal{P}$ is the set of Bernoulli (resp. Gaussian) distributions with mean in $\Theta$. As such, we take $\mathcal{Q} \supset \mathcal{P}$ so that the estimator of $P_i$ is always in $\mathcal{Q}$. In these models, we identify the distribution $P_i \in \mathcal{P}$ with its mean parameter in $\Theta \subset \mathbb{R}$.

---

[6]Section 33.3 therein.

---
**Algorithm 1:** $R^{\mathrm{go}}$-Tracking
---
**input :** ($\epsilon$-)optimal solution $(\boldsymbol{r}^*(\cdot), J^*(\cdot))$ of (2).
**1** Draw each arm once.
**2 for** $t = K + 1, 2, \ldots, T$ **do**
**3** $\quad\lfloor$ Draw arm $\arg\max_{i \in [K]} \{r_i^*(\boldsymbol{Q}(t-1)) - N_i(t-1)/(t-1)\}$.
**4 return** $J(T) = J^*(\boldsymbol{Q})$.
---

Our interest lies in the rate $\lim_{T \to \infty} \frac{1}{T} \log(1/\mathbb{P}[J(T) \notin \mathcal{I}^*(\boldsymbol{P})])$ of convergence of the PoE. Since we are interested in lower and upper bounds of the rate of algorithms including those requiring the knowledge of $T$, we define the rate for a sequence[7] of algorithms $\{\pi_T\}$ by

$$R(\{\pi_T\}) = \inf_{\boldsymbol{P} \in \mathcal{P}^K} H(\boldsymbol{P}) \liminf_{T \to \infty} \frac{1}{T} \log(1/\mathbb{P}[J(T) \notin \mathcal{I}^*(\boldsymbol{P})]). \tag{1}$$

Here, a larger $R(\{\pi_T\})$ indicates a faster convergence of the PoE.[8]

## 2.1 PoE for oracle algorithms

First, we derive a lower bound on the PoE that is unlikely to be achievable but strongly related to an optimal algorithm. Let $D(P\|Q) = \mathbb{E}_{X \sim P}[\frac{\mathrm{d}P}{\mathrm{d}Q}(X)]$ be the Kullback–Leibler (KL) divergence between $P$ and $Q$. In the case of Bernoulli distributions, $D(P\|Q) = \mu_P \log(\mu_P/\mu_Q) + (1 - \mu_P) \log((1 - \mu_P)/(1 - \mu_Q))$ where $\mu_P, \mu_Q$ are the means of $P, Q$. In the case of Gaussian distributions, $D(P\|Q) = (\mu_P - \mu_Q)^2/(2\sigma^2)$. We then have the following bound.

**Theorem 1.** Under any sequence of algorithm $\{\pi_T\}$ it holds that

$$R(\{\pi_T\}) \leq \sup_{\boldsymbol{r}(\cdot) \in \Delta_K, \, J(\cdot) \in [K]} \inf_{\boldsymbol{Q} \in \mathcal{Q}^K} \inf_{\boldsymbol{P} \in \mathcal{P}^K : J(\boldsymbol{Q}) \notin \mathcal{I}^*(\boldsymbol{P})} H(\boldsymbol{P}) \sum_{i \in [K]} r_i(\boldsymbol{Q}) D(Q_i\|P_i) =: R^{\mathrm{go}}, \tag{2}$$

where the outer supremum is taken over all functions $\boldsymbol{r}(\cdot) : \mathcal{Q}^K \to \Delta_K$, $J(\cdot) : \mathcal{Q}^K \to [K]$.

All proofs are provided in the appendix. This theorem states that under any algorithm there exists an instance $\boldsymbol{P}$ such that the PoE is at least $\exp(-TR^{\mathrm{go}}/H(\boldsymbol{P}) + o(T))$. Intuitively speaking, the bound in Theorem 1 corresponds to the best possible rate of oracle algorithms that can determine the allocation as $\boldsymbol{r} = \boldsymbol{r}^*(\boldsymbol{Q}) \in \Delta_K$ knowing the final empirical mean $\boldsymbol{Q} = \boldsymbol{Q}(T)$, where $\boldsymbol{r}^*(\cdot)$ is the ($\epsilon$-)optimal[9] solution of Eq. (2).

However, whether the rate $R^{\mathrm{go}}$ or not by some algorithm is highly nontrivial.[10] In the actual trial, the algorithm can only know the empirical mean $\boldsymbol{Q}(t-1)$ at the beginning of the current round $t$; we cannot ensure the achievability of the bound for oracle algorithms. Despite this, one reasonable choice of the algorithm would be to keep tracking this optimal allocation $\boldsymbol{r}^*(\boldsymbol{Q}(t-1))$, expecting that the current empirical mean $\boldsymbol{Q}(t-1)$ is close to $\boldsymbol{Q}(T)$. $R^{\mathrm{go}}$-tracking in Algorithm 1 is the algorithm based on this idea. Here, $N_i(t-1)$ is the number of times that the arm $i$ is drawn at the beginning of the $t$-th round, and it draws the arm such that the current fraction of the allocation $N_i(t-1)/(t-1)$ is the most insufficient compared with the estimated optimal allocation $\boldsymbol{r}^*(\boldsymbol{Q}(t-1))$.

As we will see in Section 4, the empirical performance of Algorithm 1 is very close to the PoE lower bound stated above. However, it is difficult to expect that this algorithm provably achieves this bound in general because of the following: We could prove that $R^{\mathrm{go}}$-tracking is optimal if the fraction of allocation always satisfies $N_i(t)/t = \boldsymbol{r}^*(\boldsymbol{Q}(t)) + o(1)$, that is,

---

[7] Here, we use $\{\pi_T\}$ to denote a sequence of algorithms for each $T = 1, 2, \ldots$. For example, successive halving (Audibert et al., 2010) is a sequence of algorithms in this sense.

[8] This corresponds to the rate $R = R(\{\pi_T\})$ of $\exp(-RT/H(\boldsymbol{P}))$.

[9] This paper uses $\epsilon > 0$ as an arbitrarily small gap to the optimal solution. This $\epsilon$ is introduced so that we can avoid the discussion on the existence of a supremum and does not matter much in this paper. An asterisk is used to denote optimality.

[10] We leave this an open problem.

the algorithm is able to track the ideal allocation $\boldsymbol{r}^*(\boldsymbol{Q}(t))$. However, this does not hold in general. For example, the empirical mean $\boldsymbol{Q}(t)$ sometimes changes rapidly in the Gaussian case. Whilst such an event occurs with an exponentially small probability, the PoE itself is also an exponentially small probability and it is highly nontrivial to specify in which case the tracking failure probability becomes negligible.

**Remark 1.** (Derivation of Theorem 1) From a technical point of view, the main difference from the lower bound in the FC setting is that we also have to consider candidates for empirical distributions $\boldsymbol{Q}$ and true distributions $\boldsymbol{P}$. This makes the analysis much more difficult because a slight difference of the empirical distribution might (possibly discontinuously) affect the allocation unlike the difference of the true distribution $\boldsymbol{P}$ unknown to the algorithm. A naive analysis only depending on the empirical distribution fails because of this discontinuity of the allocation. To overcome this difficulty, we adopt a technique inspired by the *typical set analysis* that is often used in the field of information theory (Cover and Thomas, 2006). We define the *typical allocation* for each candidate of empirical distribution $\boldsymbol{Q}$ and prove the theorem by evaluating the error probability based on the typical allocation.

**Remark 2.** (Trivial solutions) We can take arbitrary $H(\boldsymbol{P}) > 0$ as a complexity measure, but $R^{\mathrm{go}}$ can become zero if $H(\boldsymbol{P})$ is not taken reasonably. For example, if we take $H(\boldsymbol{P}) = \sum_i 1/\Delta_i$ (rather than $\sum_i 1/\Delta_i^2$ that is widely used), any positive rate is not achievable around a small gap $\min_i \Delta_i$, and thus $R^{\mathrm{go}} = 0$. When $R^{\mathrm{go}} = 0$ any algorithm trivially satisfies PoE $\leq \exp(-TR^{\mathrm{go}}/H(\boldsymbol{P}) + o(T))$. This means that any algorithm is minimax optimal in terms of $H(\boldsymbol{P})$, that is, such a choice of $H(\boldsymbol{P})$ gives meaningless results.

**Remark 3.** (Empirical best arm) Eq. (2) also involves the optimization of the recommendation arm $J(\boldsymbol{Q})$ and $\boldsymbol{r}(\boldsymbol{Q})$. We can easily see that it is optimal to set $J(\boldsymbol{Q}) = i^*(\boldsymbol{Q})$, that is, to take the empirical best arm as the recommendation arm when $\mathcal{P} = \mathcal{Q}$ since $R(\pi)$ otherwise becomes zero. However, $J(\boldsymbol{Q}) = i^*(\boldsymbol{Q})$ might not hold for $\boldsymbol{Q} \notin \mathcal{P}$ when $\mathcal{P} \subsetneq \mathcal{Q}$.

Sections 2.2–2.4 that follow this section are for achieving the minimax optimal rate and are of theoretical interest. A reader who is mainly interested in an actual implementation may skip these sections.

## 2.2 PoE considering trackability

To construct a provably optimal algorithm, we begin by refining the lower bound of the PoE by weakening the "strength" of the oracle algorithm.

We consider splitting $T$ rounds into $B$ batches of size $\lfloor T/B \rfloor$ or $\lfloor T/B \rfloor + 1$. Let

$$\boldsymbol{r}^B = (\boldsymbol{r}_1(\boldsymbol{Q}_1), \boldsymbol{r}_2(\boldsymbol{Q}_1, \boldsymbol{Q}_2), \boldsymbol{r}_3(\boldsymbol{Q}_1, \boldsymbol{Q}_2, \boldsymbol{Q}_3), \ldots, \boldsymbol{r}_B(\boldsymbol{Q}_1, \ldots, \boldsymbol{Q}_B))$$

be a sequence of $B$ functions, where $\boldsymbol{r}_b : \mathcal{Q}^{Kb} \to \Delta_K$ corresponds the allocation in the $b$-th batch when the empirical means of the first $b$ batches are $\boldsymbol{Q}^b = (\boldsymbol{Q}_1, \boldsymbol{Q}_2, \ldots, \boldsymbol{Q}_b)$. Based on this class of allocation rule, we have the following PoE lower bound.

**Theorem 2.** (PoE Bound for batch-oracle algorithms) Under any sequence of algorithms $\pi_T$ and $B \in \mathbb{N}$,

$$R(\{\pi_T\}) \leq \sup_{\boldsymbol{r}^B(\cdot), J(\cdot)} \inf_{\boldsymbol{Q}^B \in \mathcal{Q}^{KB}} \inf_{\boldsymbol{P}:J(\boldsymbol{Q}^B) \notin \mathcal{I}^*(\boldsymbol{P})} \frac{H(\boldsymbol{P})}{B} \sum_{i \in [K], b \in [B]} r_{b,i} D(Q_{b,i} \| P_i) =: R_B^{\mathrm{go}}. \quad (3)$$

Here, the outer supremum is taken over all functions $\boldsymbol{r}^B(\cdot) = (\boldsymbol{r}_1(\cdot), \boldsymbol{r}_2(\cdot), \ldots, \boldsymbol{r}_B(\cdot))$ for $\boldsymbol{r}_b(\cdot) : \mathcal{Q}^{Kb} \to \Delta_K$ and $J(\cdot) : \mathcal{Q}^{KB} \to [K]$.

Theorem 1 is the special case of this theorem with $B = 1$. This bound corresponds to the best bound of oracle algorithms that can determine the allocation of the $b$-th batch knowing the empirical distribution of this batch. It is tighter than that given in Theorem 1, as the oracle considered here cannot know the empirical distribution of the later batches $b+1, b+2, \ldots, B$. It follows that we can obtain the following result.

**Corollary 3.** We have $R_B^{\mathrm{go}} \leq R^{\mathrm{go}}$ for any $B \in \mathbb{N}$.

We will show that $R_\infty^{\mathrm{go}} := \lim_{B \to \infty} R_B^{\mathrm{go}}$ exists and is the best possible rate.

**Algorithm 2:** Delayed optimal tracking (DOT)

---

**input :** $\epsilon$-optimal solution $\boldsymbol{r}^{B,*}(\cdot) = (\boldsymbol{r}_1^*(\cdot), \boldsymbol{r}_2^*(\cdot), \ldots, \boldsymbol{r}_B^*(\cdot), J^*(\cdot))$ of Eq. (3).

**1 for** $b = 1, 2, \ldots, K$ **do**
**2** $\quad$ Set $r_{b,i} = \mathbf{1}[i = b]$ for $i \in [K]$ and draw arm $b$ for $T_B$ times.
**3** Set $\boldsymbol{Q}_1' := \boldsymbol{Q}_K$ for the empirical mean $\boldsymbol{Q}_K$.
**4 for** $b = K + 1, K + 2, \ldots, B + K - 1$ **do**
**5** $\quad$ Compute $\boldsymbol{r}_b = (r_{b,1}, r_{b,2}, \ldots, r_{b,K}) = \boldsymbol{r}_{b-K}^*(\boldsymbol{Q}_1', \boldsymbol{Q}_2', \ldots, \boldsymbol{Q}_{b-K}')$.
**6** $\quad$ Draw each arm $i$ for $n_{b,i}$ times, where $n_{b,i} \geq r_{b,i}(T_B - K)$ is taken so that
$\quad\quad \sum_{i \in [K]} n_{b,i} = T_B$.
**7** $\quad$ Observe empirical mean $\boldsymbol{Q}_b$ of the batch.
**8** $\quad$ Update the *stored* empirical average as

$$\boldsymbol{Q}_{b-K+1}' = \boldsymbol{Q}_{b-K}' + \boldsymbol{r}_b(\boldsymbol{Q}_b - \boldsymbol{Q}_{b-K}'),$$

$\quad\quad$ where $\boldsymbol{r}_b\boldsymbol{Q}$ denotes the element-wise product.
**9** Recommend $J(T) = J^*(\boldsymbol{Q}_1', \boldsymbol{Q}_2', \ldots, \boldsymbol{Q}_B')$.

---

### 2.3 Matching algorithm

In this section, we derive an algorithm that has a rate that almost matches $R_B^{\mathrm{go}}$. For any $\epsilon > 0$, let an ($\epsilon$-)optimal solution[11] of Eq. (3) be $(\boldsymbol{r}^{B,*}(\cdot), J^*(\cdot)) = (\boldsymbol{r}_1^*(\cdot), \boldsymbol{r}_2^*(\cdot), \boldsymbol{r}_3^*(\cdot), \ldots, \boldsymbol{r}_B^*(\cdot), J^*(\cdot))$ with its objective at least

$$\inf_{\boldsymbol{Q}^B \in \mathcal{Q}^{KB}} \inf_{\boldsymbol{P}:J^*(\boldsymbol{Q}^B) \notin \mathcal{I}^*(\boldsymbol{P})} \frac{H(\boldsymbol{P})}{B} \sum_{i \in [K], b \in [B]} r_{b,i}^*(\boldsymbol{Q}^b) D(Q_{b,i} \| P_i) \geq R_B^{\mathrm{go}} - \epsilon.$$

We cannot naively follow the allocation $r_{b,i}^*(\boldsymbol{Q}^b)$ because it requires the empirical mean of the current batch $\boldsymbol{Q}_b$, which is not fully available until the end of the current batch. The delayed optimal tracking algorithm (DOT, Algorithm 2) addresses this issue. This algorithm divides $T$ rounds into $B + K - 1$ batches, where the $b$-th batch corresponds to $(bT_B + 1, bT_B + 2, \ldots, (b+1)T_B)$-th rounds for $T_B = T/(B + K - 1)$. Here, for simplicity, we assume that $T$ is a multiple of $B + K - 1$. In the other case, we can reach almost the same result by just ignoring the last $T - (B + K - 1)\lfloor T/(B + K - 1) \rfloor$ rounds.

The crux of Algorithm 2 is to determine allocation $\boldsymbol{r}_b$ by using the *stored* empirical mean $\boldsymbol{Q}_1', \boldsymbol{Q}_2', \ldots \boldsymbol{Q}_B'$ rather than the true empirical mean $\boldsymbol{Q}_1, \boldsymbol{Q}_2, \ldots \boldsymbol{Q}_{B+K-1}$: The first $K$ batches are devoted to uniform exploration and the samples are stored in a queue (though this explanation is not strict, in that the actual procedure is done after taking the mean of the stored samples). At the $b$-th batch for $b \geq K + 1$, we draw each arm $i$ for $n_{b,i} \approx T_B r_{b,i}$ times[12], where $\boldsymbol{r}_b = (r_{b,i})_{i \in [K]}$ is determined based on the stored samples in the queue. When drawing arm $i$ for $n_{b,i}$ times, we dequeue and open $n_{b,i}$ stored samples instead of opening actual $n_{b,i}$ samples, the latter of which are enqueued and kept unopened.[13]

By the nature of this algorithm, we can ensure the following property.

**Lemma 4.** Assume that we run Algorithm 2. Then, the following inequality always holds:

$$\frac{1}{B + K - 1} \sum_{i \in [K], b \in [B+K-1]} r_{b,i} D(Q_{b,i} \| P_i) \geq \frac{B}{B + K - 1} \frac{R_B^{\mathrm{go}} - \epsilon}{H(\boldsymbol{P})}. \tag{4}$$

Lemma 4 states that the empirical divergence of DOT given in the LHS of Eq. (4) almost matches the upper bound $R_B^{\mathrm{go}}/H(\boldsymbol{P})$ for sufficiently large $B$ despite delayed allocation. Using this property, we obtain the following bound.

---

[11] Again, $\epsilon > 0$ can be arbitrarily small and is introduced to avoid the issue on the existence of the supremum.

[12] The $-K$ in Line 6 of Algorithm 2 is for the ceiling fractional values. This is reflected in the term $T'$ in Theorem 5. If $T$ is large compared to $B, K$, the difference between $T$ and $T'$ does not matter.

[13] In other words, observations do not affect the allocation until opened.

**Theorem 5.** (Performance bound of Algorithm 2) The PoE of the DOT algorithm satisfies

$$\mathbb{P}[J(T) \notin \mathcal{I}^*(\boldsymbol{P})] \le \exp\left(-\frac{BT'}{B+K-1}\frac{R_B^{\mathrm{go}}-\epsilon}{H(\boldsymbol{P})} + f(K,B,T)\right),$$

where $T' = T - (B+K-1)K$ and $f(K,B,T) = 2BK\log(2T)$.

The following corollary is immediate since $f(K,B,T) = o(T)$ holds for fixed $K, B$.

**Corollary 6.** The worst-case rate of the DOT algorithm $\pi_{\mathrm{DOT},T}$ satisfies

$$R(\{\pi_{\mathrm{DOT},T}\}) \ge \frac{B}{B+K-1}(R_B^{\mathrm{go}} - \epsilon).$$

### 2.4 Optimality

In this section, we show that the rate $R(\pi_{\mathrm{DOT}})$ of DOT becomes arbitrarily close to optimal when we take a sufficiently large number of batches $B$.

**Theorem 7.** (Optimality of DOT) Assume $H(\boldsymbol{P})$ be such that $R^{\mathrm{go}} < \infty$. Then, the limit

$$R_\infty^{\mathrm{go}} := \lim_{B\to\infty} R_B^{\mathrm{go}} \tag{5}$$

exists. Moreover, for any $\eta > 0$, there exist parameters $B, \epsilon$ such that the following holds on the performance of the DOT algorithm:

$$R(\{\pi_{\mathrm{DOT},T}\}) = \inf_{\boldsymbol{P}\in\mathcal{P}} H(\boldsymbol{P}) \liminf_{T\to\infty} \frac{\log(1/\mathbb{P}[J(T) \notin \mathcal{I}^*(\boldsymbol{P})])}{T} \ge R_\infty^{\mathrm{go}} - \eta. \tag{6}$$

**Remark 4.** ($\eta$-optimality) Since $R(\{\pi_T\}) \le R_B^{\mathrm{go}}$ holds for any sequence of algorithms $\{\pi_T\}$ and $B \in \mathbb{N}$, we have

$$R(\{\pi_T\}) \le \inf_{B\in\mathbb{N}} R_B^{\mathrm{go}} \le \liminf_{B\to\infty} R_B^{\mathrm{go}} = R_\infty^{\mathrm{go}}$$

from Eq. (5).

Remark 4 states that the rate of the DOT algorithm given in Eq. (6) is optimal up to $\eta$ for arbitrary small $\eta > 0$. This essentially states that, no algorithm can be $\eta$-better than DOT in terms of the rate against the worst-case instance $\boldsymbol{P}$. In that sense, DOT is asymptotically minimax optimal. Note that the discussion here first takes $T \to \infty$, and then takes a large $B$, which implies that $B$ should be taken as $o(T)$ (i.e., each batch has a sufficiently large number of samples).

**Remark 5.** (Utility of the DOT algorithm) Although DOT (Algorithm 2) has an asymptotically optimal rate $R_\infty^{\mathrm{go}}$, it is difficult to calculate, or to even approximate, the optimal solution of Eq. (3) since it is not an optimization of a finite-dimensional vector but an optimization of function $\boldsymbol{r}^B$, which has a high input dimension proportional to $B$. In this sense, the DOT algorithm as well as Theorem 7 is purely theoretical thus far, and the existence of a computationally tractable and provably optimal algorithm is an important open question.

Instead of pursuing the direction of batching, the subsequent sections rather focus on implementing Algorithm 1.

## 3 Learning

In this section, we propose a method to learn $\boldsymbol{r}(\boldsymbol{Q})$ of Eq. (2) by utilizing a neural network to practically realize $R^{\mathrm{go}}$-tracking in Algorithm 1. Throughout this section, we assume a class of algorithms that recommend the empirical best arm, which is guaranteed to be optimal when $\mathcal{P} = \mathcal{Q}$ (see Remark 3). This section focuses on Bernoulli arms, and with a slight abuse of notation, we use $P_i$ and $Q_i$ to denote the true mean and the empirical mean of arm $i$, respectively.

**Algorithm 3:** Gradient descent method for $\boldsymbol{\theta}$

**input :** learning rate $\eta$

**1 while** *not converged* **do**

    /* Compute $\boldsymbol{P}^{\min}$ and $\boldsymbol{Q}^{\min}$ which minimizes the negative exponent */

**2**    Set $E^{\min} \leftarrow \infty$.

**3**    **for** $n_1 = 1, 2, \ldots, N^{\text{true}}$ **do**

**4**        Sample true parameters $\boldsymbol{P}$ from $\mathcal{P}$ uniformly at random.

**5**        **for** $n_2 = 1, 2, \ldots, N^{\text{emp}}$ **do**

**6**            Sample $\boldsymbol{Q}$ from $\{\boldsymbol{Q} \in \mathcal{Q}^K : \mathcal{I}^*(\boldsymbol{Q}) \cap \mathcal{I}^*(\boldsymbol{P}) = \emptyset\}$.

**7**            **if** $E(\boldsymbol{P}, \boldsymbol{Q}; \boldsymbol{\theta}) < E^{\min}$ **then**

**8**                $\boldsymbol{P}^{\min} \leftarrow \boldsymbol{P}, \quad \boldsymbol{Q}^{\min} \leftarrow \boldsymbol{Q}, \quad E^{\min} \leftarrow E(\boldsymbol{P}^{\min}, \boldsymbol{Q}^{\min}; \boldsymbol{\theta})$

**9**    Update parameters $\boldsymbol{\theta} \leftarrow \boldsymbol{\theta} - \eta \nabla_{\boldsymbol{\theta}} E(\boldsymbol{P}^{\min}, \boldsymbol{Q}^{\min}; \boldsymbol{\theta})$.

---

**Algorithm 4:** $R^{\text{go}}$-Tracking by Neural Network (TNN)

**1** Draw each arm once.

**2 for** $t = K + 1, 2, \ldots, T$ **do**

**3**    Draw arm $\arg\max \left( r_{\boldsymbol{\theta},i}(\boldsymbol{Q}(t-1)) - N_i(t-1)/(t-1) \right)$.

**4 return** $J(T) = \arg\max Q_i(T)$ (empirical best arm).

---

### 3.1 Learning allocation

Let $\boldsymbol{r_\theta}(\boldsymbol{Q}) : \mathcal{Q}^K \to \Delta_K$ be a neural network with a set of parameters $\boldsymbol{\theta}$. We consider alternately optimizing $r_{\boldsymbol{\theta}}(\cdot)$ and the worst pair of instances $(\boldsymbol{P}, \boldsymbol{Q})$, and we update $\boldsymbol{\theta}$ via mini-batch gradient descent. Given a complexity function $H(\boldsymbol{P})$, Eq. (2) is defined as the minimum over all $(\boldsymbol{P}, \boldsymbol{Q})$ such that the best arm is different. Our learning method (Algorithm 3) uses $L$ mini-batches. Let

$$E(\boldsymbol{P}, \boldsymbol{Q}; \boldsymbol{\theta}) \coloneqq H(\boldsymbol{P}) \sum_{i=1}^{K} r_{\boldsymbol{\theta},i}(\boldsymbol{Q}) D\left(Q_i \| P_i\right). \tag{7}$$

Given allocation $\boldsymbol{r_\theta}$, Eq. (7) is the negative log-likelihood (rate) of the bandit instance $\boldsymbol{P}$ against the empirical means $\boldsymbol{Q}$. At each batch, it obtains the pair $\boldsymbol{P}^{\min}, \boldsymbol{Q}^{\min}$ such that Eq. (7) is minimized. Specifically, for each iteration, we sample $N^{\text{true}}$ candidates of true means $\boldsymbol{P}$ uniformly from $\mathcal{P}^K$, then for each $\boldsymbol{P}$, we sample $N^{\text{emp}}$ values of empirical means $\boldsymbol{Q} \in \mathcal{Q}^K$ such that $\mathcal{I}^*(\boldsymbol{Q}) \cap \mathcal{I}^*(\boldsymbol{P}) = \emptyset$ uniformly at random.

### 3.2 Tracking by neural network

Having trained $\boldsymbol{r_\theta}$, we propose the $R^{\text{go}}$-Tracking by the neural network (TNN) algorithm (Algorithm 4), which is an implementation of $R^{\text{go}}$-Tracking by the trained neural network. This algorithm draws the arm such that the current fraction of samples $N_i(t-1)/(t-1)$ is the most insufficient compared to the learned allocation $\boldsymbol{r_\theta}(\boldsymbol{Q}(t-1))$.

## 4 Simulation

This section numerically tests the performance of the TNN algorithm.[14] We compared the performance of TNN (Algorithm 4) with two algorithms: Uniform algorithm, which samples each arm in a round-robin fashion, and Successive Rejects (SR, Audibert et al., 2010), where the entire trial is divided into segments before the game starts, and one arm with the smallest estimated mean reward is removed for each segment.

---

[14]The source code of the simulation is available at `https://github.com/tsuchhiii/fixed-budget-bai`.

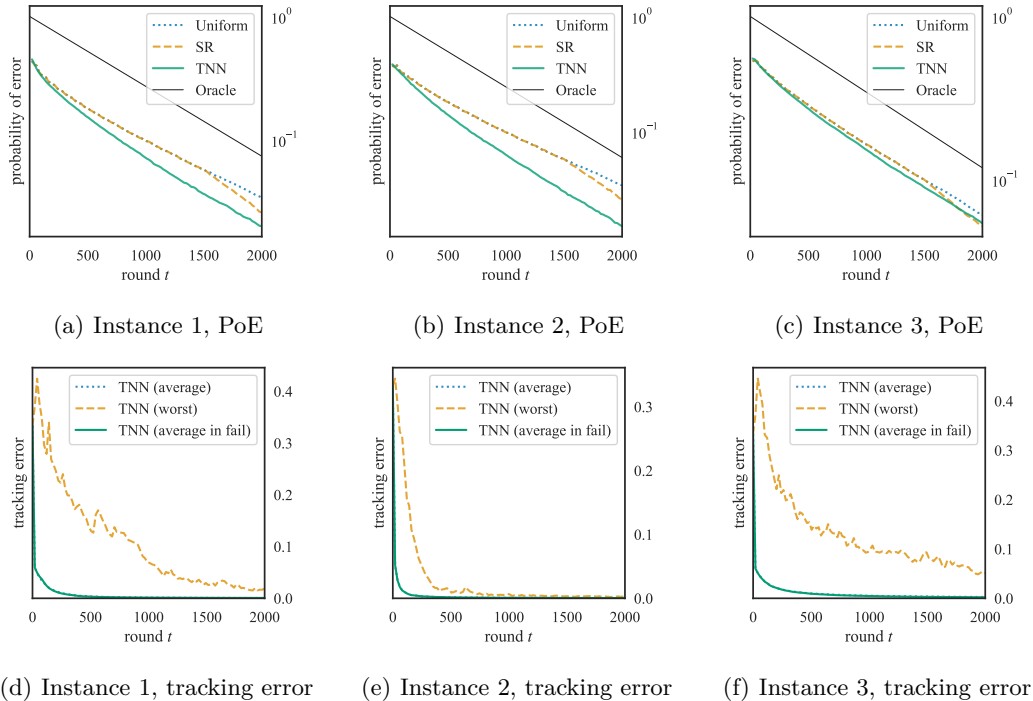



(a) Instance 1, PoE     (b) Instance 2, PoE     (c) Instance 3, PoE

(d) Instance 1, tracking error    (e) Instance 2, tracking error    (f) Instance 3, tracking error



Figure 1: Bernoulli bandits, $K = 3, T = 2000$, average over $10^5$ trials.

We consider Bernoulli bandits with $K = 3$ arms, where each mean parameter is in $[0, 1]$. In particular, we consider the three sets of true parameters: (instance 1) $\boldsymbol{P} = (0.5, 0.45, 0.3)$, (instance 2) $\boldsymbol{P} = (0.5, 0.45, 0.05)$, and (instance 3) $\boldsymbol{P} = (0.5, 0.45, 0.45)$. The number of the rounds $T$ is fixed to 2000, and we repeated the experiments for $10^5$ times.

## 4.1 Training neural networks

Here, we show experimental details for training neural networks for the TNN algorithm discussed in Section 3.2.

We used the complexity measure $H_1(\boldsymbol{P}) = \sum_{i \neq i^*(\boldsymbol{P})} (P^* - P_i)^{-2}$ as a standard choice of $H(\boldsymbol{P})$. We used the neural network with four layers (including the input layer and output layer), where we used the ReLU for the activation functions and introduced the skip-connection (He et al., 2016) between each hidden layer to make training the network easier. To obtain the map to $\Delta_K$, we adopted the softmax function. The number of nodes in the hidden layers was fixed to $K \times 3$. We used AdamW (Loshchilov and Hutter, 2019) with a learning rate $10^{-3}$ and weight decay $10^{-7}$ to update the parameters.

For training the neural network, we ran Algorithm 3 with $N^{\text{true}} = 32$ and $N^{\text{emp}} = 90$. Furthermore, to allow the neural network to easily learn $\boldsymbol{r}$, the elements of $\boldsymbol{P} = (P_1, P_2, \ldots, P_K)$ were sorted beforehand.

## 4.2 Experimental results

Figure 1 illustrates the results of our simulations. Each column corresponds to the result for each instance.

The first row ((a)–(c)) shows the PoE of the compared methods when the arm with the largest empirical mean is regarded as the estimated best arm $J(t)$ at each round $t$. Here, the black line represents $\exp(-t \inf_{\boldsymbol{Q}} \sum_i r_{\boldsymbol{\theta}, i}(\boldsymbol{Q}) D(Q_i \| P_i))$, which corresponds to the exponent of the oracle algorithm that can perfectly track the allocation $r_{\boldsymbol{\theta}, i}(\boldsymbol{Q})$. Therefore, the asymptotic

slope of TNN cannot be better than that of the black line. We can see from the figures that the slope of the TNN is close to the oracle algorithm and performs better than or comparable to the other algorithms. Note that this is the result for fixed time horizon $T$. Although the final slope of SR seems to be steeper than that of TNN, it is just due to the fact that SR is not anytime and is an algorithm that divides $T$ rounds into several segments.

The second row ((d)–(f)) shows the tracking error of the TNN algorithm, which is defined as $\mathrm{disc}(t) = \max_{i \in [K]} |r_i(\boldsymbol{Q}(t)) - N_i(t)/t|$, which measures the discrepancy between the ideal allocation $r_i(\boldsymbol{Q}(t))$ and the actual allocation $N_i(t)/t$. If this quantity is $o(T)$ in almost all tests (including those in which the algorithm failed to recommend the best arm) and in all instances, then we can guarantee $R^{\mathrm{go}} = R^{\mathrm{go}}_{\infty}$. The labels TNN (average), TNN (worst), and TNN (average in fail) correspond to the average tracking error of all trials, the worst-case tracking error and the average tracking error of all failed trials, respectively. The fact that 'TNN (worst)' is small at $T = 2{,}000$ implies that the gap between $R^{\mathrm{go}}$ and $R^{\mathrm{go}}_{\infty}$ is small, which supports the reasonableness of algorithms based on $R^{\mathrm{go}}$.

## 5   Conclusion

This paper considered the fixed-budget best arm identification problem. We identified the minimax rate $R^{\mathrm{go}}_{\infty}$ on the exponent of the probability of error by introducing a matching algorithm (DOT algorithm). Optimization of rate $R^{\mathrm{go}}_{\infty}$ is very challenging to implement, and we considered learning a simpler optimization problem of rate $R^{\mathrm{go}}$ by using a neural network (TNN algorithm). The TNN algorithm outperformed existing algorithms. Several possible lines of future work include the following points.

- More scalable learning of $\boldsymbol{r}(\boldsymbol{Q})$: TNN adopted a neural network to obtain the oracle allocation $\boldsymbol{r}(\boldsymbol{Q})$. While its empirical results are promising and support our theoretical findings, the current experiment is limited to the case of $K = 3$ arms because the learning is very costly even for small $K$. A more sophisticated learning algorithm is desired to realize $R^{\mathrm{go}}$-tracking for larger $K$.

- Learning a complexity measure $H(\boldsymbol{P})$: We have assumed the complexity measure $H(\boldsymbol{P})$ is given exogenously. A principled way to choose $H(\boldsymbol{P})$ is an interesting future work.

- Identifying the existence (or nonexistence) of the gap: Although the empirical results suggest that $R^{\mathrm{go}}$ is very close (or maybe equal) to the optimal rate $R^{\mathrm{go}}_{\infty}$ for the Bernoulli case, a formal analysis of this gap for general cases is demanded since the DOT algorithm to achieve $R^{\mathrm{go}}_{\infty}$ is computationally almost infeasible.

- A bound for another rate measure: we defined the worst-case rate by (1), which first takes the limit of $T$ and then takes the worst-case instance $\boldsymbol{P}$. Another natural choice of the rate would be to exchange them, that is, to consider

$$R'(\{\pi_T\}) = \liminf_{T \to \infty} \inf_{\boldsymbol{P} \in \mathcal{P}^K} \frac{H(\boldsymbol{P})}{T} \log(1/\mathbb{P}[J(T) \notin \mathcal{I}^*(\boldsymbol{P})]) \leq R(\{\pi_T\}).$$

Whereas Theorems 1 and 2 on the upper bounds of $R(\pi)$ are still valid for $R'(\{\pi_T\}) \leq R(\{\pi_T\})$, the current achievability analysis does not apply and analyzing the tightness of $R^{\mathrm{go}}_{\infty}$ for $R'(\{\pi_T\})$ is an open problem.

## 6   Acknowledgement

The authors thank the reviewers and the associated editor for constructive discussion and suggested corrections. To include the discussion during the review process, we have rewritten Sections B and G.1. We have also added Section C. All errors in the paper are our own. TT was supported by JST, ACT-X Grant Number JPMJAX210E, Japan and JSPS, KAKENHI Grant Number JP21J21272, Japan. JH was supported by JSPS KAKENHI Grant Number JP21K11747.

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
