Table 1: Major notation

| symbol | definition |
| --- | --- |
| $K$ | number of the arms |
| $T$ | number of the rounds |
| $B$ | number of the batches |
| $T_B$ | $= T/(B + K - 1)$ |
| $T'$ | $= T - (B + K - 1)K$ |
| $I(t)$ | arm selected at round $t$ |
| $X(t)$ | reward at round $t$ |
| $J(T)$ | recommendation arm at the end of round $T$ |
| $\mathcal{P}$ | hypothesis class of $\boldsymbol{P}$ |
| $\mathcal{Q}$ | distribution of estimated parameter of $\boldsymbol{Q}$ |
| $\boldsymbol{P} \in \mathcal{P}^K$ | true parameters |
| $P_i \in \mathcal{P}$ | $i$-th component of $\boldsymbol{P}$ |
| $\mathcal{I}^* = \mathcal{I}^*(\boldsymbol{P})$ | set of best arms under parameter $\boldsymbol{P}$ |
| $i^*(\boldsymbol{P})$ | one arm in $\mathcal{I}^*(\boldsymbol{P})$ (taken arbitrary in a deterministic way) |
| $\boldsymbol{Q} \in \mathcal{Q}^K$ | estimated parameters of $\boldsymbol{P}$ |
| $Q_i \in \mathcal{Q}$ | $i$-th component of $\boldsymbol{Q}$ |
| $\boldsymbol{Q}_b \in \mathcal{Q}^K$ | estimated parameters of $b$-th batch |
| $Q_{b,i} \in \mathcal{Q}$ | $i$-th component of $\boldsymbol{Q}_b$ |
| $\boldsymbol{Q}^b \in \mathcal{Q}^{Kb}$ | $= (\boldsymbol{Q}_1, \boldsymbol{Q}_2, \ldots, \boldsymbol{Q}_b)$ |
| $\boldsymbol{Q}'_b \in \mathcal{Q}^K$ | stored parameters (in Algorithm 2) |
| $Q'_{b,i} \in \mathcal{Q}$ | $i$-th component of $\boldsymbol{Q}'_b$ |
| $D(Q\|P)$ | KL divergence between $Q$ and $P$ |
| $\Delta_K$ | probability simplex in $K$ dimensions |
| $\boldsymbol{r} \in \Delta_K$ | allocation (proportion of arm draws) |
| $r_i$ | $i$-th component of $\boldsymbol{r}$ |
| $\boldsymbol{r}_b \in \Delta_K$ | allocation at $b$-th batch |
| $r_{b,i}$ | $i$-th component of $\boldsymbol{r}_b$ |
| $\boldsymbol{r}^b$ | $= (\boldsymbol{r}_1, \boldsymbol{r}_2, \ldots, \boldsymbol{r}_b)$ |
| $\boldsymbol{n}_b$ | numbers of draws of Algorithm 2 at $b$-th batch |
| $n_{b,i}$ | $i$-th component of $\boldsymbol{n}_b$. Note that $n_{b,i} \geq r_{b,i}(T_B - K)$ holds. |
| $J(\boldsymbol{Q}^B)$ | recommendation arm given $\boldsymbol{Q}^B$ |
| $(\boldsymbol{r}^{B,*}, J^*)$ | $\epsilon$-optimal allocation |
| $H(\cdot)$ | complexity measure of instances |
| $R(\{\pi_T\})$ | worst-case rate of PoE of sequence of algorithms $\{\pi_T\}$ in (1) |
| $R^{\text{go}}$ | best possible $R(\{\pi_T\})$ for oracle algorithms in (2) |
| $R_B^{\text{go}}$ | best possible $R(\{\pi_T\})$ for $B$-batch oracle algorithms in (3) |
| $R_\infty^{\text{go}}$ | $\lim_{B \to \infty} R_B^{\text{go}}$. Limit exists (Theorem 7) |
| $\boldsymbol{\theta}$ | model parameter of the neural network |
| $\boldsymbol{r}_{\boldsymbol{\theta}}$ | allocation by a neural network with model parameters $\boldsymbol{\theta}$ |
| $r_{\boldsymbol{\theta},i}$ | $i$-th component of $\boldsymbol{r}_{\boldsymbol{\theta}}$ |

# A   Notation table

Table 1 summarizes our notation.

# B   Uniform optimality in the fixed-confidence setting

For sufficiently small $\delta > 0$, the asymptotic sample complexity for the FC setting is known.

Namely, any fixed-confidence $\delta$-PAC algorithm require at least $C^{\mathrm{conf}}(\boldsymbol{P})\log\delta^{-1}+o(\log\delta^{-1})$ samples, where

$$C^{\mathrm{conf}}(\boldsymbol{P})=\left(\sup_{\boldsymbol{r}(\boldsymbol{P})\in\Delta_K}\inf_{\boldsymbol{P}':i^*(\boldsymbol{P}')\notin\mathcal{I}^*(\boldsymbol{P})}\sum_{i=1}^{K}r_iD(P_i\|P_i')\right)^{-1}. \tag{8}$$

Garivier and Kaufmann (2016) proposed *C*-Tracking and *D*-Tracking algorithms that have a sample complexity bound that matches Eq. (8). This achievability bound implies that there is no tradeoff between the performances for different instances $\boldsymbol{P}$, and sacrificing the performance for some $\boldsymbol{P}$ never improves the performance for another $\boldsymbol{P}'$. To be more specific, for example, even if we consider a ($\delta$-correct) algorithm that has a suboptimal sample complexity of $2C^{\mathrm{conf}}(\boldsymbol{P})\log\delta^{-1}+o(\log\delta^{-1})$ for some instance $\boldsymbol{P}$, it is still impossible to achieve sample complexity better than $C^{\mathrm{conf}}(\boldsymbol{Q})\log\delta^{-1}+o(\log\delta^{-1})$ for another instance $\boldsymbol{P}'$ as far as the algorithm is $\delta$-PAC.

## C  Suboptimal performance of fixed-confidence algorithms in view of fixed-budget setting

This section shows that an optimal algorithm for the FC-BAI can be arbitrarily bad for the FB-BAI.

For a small $\epsilon\in(0,0.1)$, consider a three-armed Bernoulli bandit instance with $\boldsymbol{P}^{(1)}=(0.6,0.5,0.5-\epsilon)$ and $\boldsymbol{P}^{(2)}=(0.4,0.5,0.5-\epsilon)$. Here, the best arm is arm 1 (resp. arm 2) in the instance $\boldsymbol{P}^{(1)}$ (resp. $\boldsymbol{P}^{(2)}$).

Let $\boldsymbol{r}^{\mathrm{conf}}(\boldsymbol{P})=(r_1^{\mathrm{conf}}(\boldsymbol{P}),r_2^{\mathrm{conf}}(\boldsymbol{P}),r_3^{\mathrm{conf}}(\boldsymbol{P}))$ be the optimal FC allocation of Eq. (8). The following characterizes the optimal allocation for $\boldsymbol{P}^{(1)},\boldsymbol{P}^{(2)}$:

**Lemma 8.** The optimal solution of Eq. (8) for instance $\boldsymbol{P}^{(1)}$ satisfies the following:

$$r_1^{\mathrm{conf}}(\boldsymbol{P}^{(1)}),r_2^{\mathrm{conf}}(\boldsymbol{P}^{(1)}),r_3^{\mathrm{conf}}(\boldsymbol{P}^{(1)})\geq 0.07=\Theta(1).$$

**Lemma 9.** The optimal solution of Eq. (8) for instance $\boldsymbol{P}^{(2)}$ satisfies the following:

$$r_1^{\mathrm{conf}}(\boldsymbol{P}^{(2)}),r_2^{\mathrm{conf}}(\boldsymbol{P}^{(2)}),r_3^{\mathrm{conf}}(\boldsymbol{P}^{(2)})=\Theta(\epsilon^2),\Theta(1),\Theta(1).$$

These two lemmas are derived in Section C.1.

Assume that we run an FC algorithm that draws arms according to allocation $\boldsymbol{r}^{\mathrm{conf}}(\cdot)$ in an FB problem with $T$ rounds. Under the parameters $\boldsymbol{P}^{(2)}$, it draws arm 1 for $O(\epsilon^2)+o(T)$ times. Letting $\delta=\boldsymbol{P}^{(1)}[J(T)=2]$, Lemma 1 in Kaufmann et al. (2016) implies that

$$(TO(\epsilon^2)+o(T))D(0.4\|0.6)\geq d(\boldsymbol{P}^{(2)}[J(T)=2],\boldsymbol{P}^{(1)}[J(T)=2])$$
$$\geq d(1/2,\boldsymbol{P}^{(1)}[J(T)=2]) \quad\text{(assuming the consistency of algorithm)}$$

$$=\frac{1}{2}\left(\log\left(\frac{1}{2\delta}\right)+\log\left(\frac{1}{2(1-\delta)}\right)\right)$$
$$\geq\frac{1}{2}\log\left(\frac{1}{2\delta}\right),$$

which implies

$$\boldsymbol{P}^{(1)}[J(T)=2]=\delta\geq\frac{1}{2}\exp\left(-2\left(TO(\epsilon^2)+o(T)\right)D(0.4\|0.6)\right). \tag{9}$$

The exponent of Eq.(9) can be arbitrarily small as $\epsilon\to+0$. In other words, the rate of this algorithm can be arbitrarily close to 0, while the complexity is $H_1(\boldsymbol{P}^{(1)})=\Theta(1)$. This fact implies that the optimal algorithm for the FC-BAI has an arbitrarily bad performance in terms of the minimax rate of the FB-BAI.

### C.1 Proofs of Lemmas 8 and 9

*Proof of Lemma 8.* For $\boldsymbol{r} = (1/3, 1/3, 1/3)$, we have

$$\inf_{\boldsymbol{P}':i^*(\boldsymbol{P}')\notin \mathcal{I}^*(\boldsymbol{P}^{(1)})} \sum_{i=1}^{K} r_i D(P_i^{(1)}\|P_i') > \frac{1}{3} \min\left(D(0.6\|0.55), D(0.5\|0.55)\right)$$

$$\text{(by } i^*(\boldsymbol{P}') \notin \mathcal{I}^*(\boldsymbol{P}^{(1)}) \text{ implies } P_1' < 0.55 \text{ or } P_2' > 0.55 \text{ or } P_3' > 0.55)$$

$$\geq 1/600.$$

We have

$$\inf_{\boldsymbol{P}':i^*(\boldsymbol{P}')\notin \mathcal{I}^*(\boldsymbol{P})} \sum_{i=1}^{K} r_1^{\mathrm{conf}}(\boldsymbol{P}^{(1)}) D(P_i\|P_i') \leq r_1^{\mathrm{conf}}(\boldsymbol{P}^{(1)}) D(0.6\|0.5)$$

$$\text{(on instance } \boldsymbol{P}' = (0.5, 0.5, 0.5 - \epsilon))$$

$$\leq 0.021 r_1,$$

which implies $r_1^{\mathrm{conf}}(\boldsymbol{P}^{(1)}) \geq (1/600) \times (1/0.021) \geq 0.07$ for the optimal allocation $r_1^{\mathrm{conf}}(\boldsymbol{P}^{(1)})$. Similar discussion yields $r_2, r_3 \geq 0.07$. □

*Proof of Lemma 9.* For $\boldsymbol{r} = (1/3, 1/3, 1/3)$, we have

$$\inf_{\boldsymbol{P}':i^*(\boldsymbol{P}')\notin \mathcal{I}^*(\boldsymbol{P}^{(2)})} \sum_{i=1}^{K} r_i D(P_i^{(2)}\|P_i')$$

$$> \frac{1}{3} \min\left(D(0.5\|0.5 - \epsilon/2), D(0.5 - \epsilon\|0.5 - \epsilon/2)\right),$$

$$\text{(by } \boldsymbol{P}' \notin \mathcal{I}^*(\boldsymbol{P}^{(2)}) \text{ implies } P_2' < 0.5 - \epsilon/2 \text{ or } P_1' > 0.5 - \epsilon/2 \text{ or } P_3' > 0.5 - \epsilon/2)$$

$$\geq \frac{\epsilon^2}{6}.$$

$$\text{(by Pinsker's inequality)}$$

We have

$$\inf_{\boldsymbol{P}':i^*(\boldsymbol{P}')\notin \mathcal{I}^*(\boldsymbol{P}^{(2)})} \sum_{i=1}^{K} r_i^{\mathrm{conf}}(\boldsymbol{P}^{(2)}) D(P_i^{(2)}\|P_i') \leq r_2^{\mathrm{conf}}(\boldsymbol{P}^{(2)}) D(0.5\|0.5 - \epsilon/2),$$

$$\text{(on instance } \boldsymbol{P}' = (0.4, 0.5 - \epsilon/2, 0.5 - \epsilon/2))$$

which implies $r_i^{\mathrm{conf}}(\boldsymbol{P}^{(2)}) = \Omega(1)$ for the optimal allocation. Similar discussion yields $r_3^{\mathrm{conf}}(\boldsymbol{P}^{(2)}) = \Omega(1)$.

In the rest of this proof, we show $r_1^{\mathrm{conf}}(\boldsymbol{P}^{(2)}) = O(\epsilon^2)$. For the ease of exposition, we drop $(\boldsymbol{P}^{(2)})$ to denote $\boldsymbol{r}^{\mathrm{conf}} = (r_1^{\mathrm{conf}}, r_2^{\mathrm{conf}}, r_3^{\mathrm{conf}})$. Lemma 4 in Garivier and Kaufmann (2016) states that the optimal solution satisfies:

$$(r_2^{\mathrm{conf}} + r_1^{\mathrm{conf}})I_{\frac{r_2^{\mathrm{conf}}}{r_2^{\mathrm{conf}}+r_1^{\mathrm{conf}}}}(P_2^{(2)}, P_1^{(2)}) = (r_2^{\mathrm{conf}} + r_3^{\mathrm{conf}})I_{\frac{r_2^{\mathrm{conf}}}{r_2^{\mathrm{conf}}+r_3^{\mathrm{conf}}}}(P_2^{(2)}, P_3^{(2)}), \qquad (10)$$

where

$$I_\alpha(P_2^{(2)}, P_i^{(2)}) = \alpha D\left(P_2^{(2)}, \alpha P_2^{(2)} + (1-\alpha)P_i^{(2)}\right) + (1-\alpha)D\left(P_i^{(2)}, \alpha P_2^{(2)} + (1-\alpha)P_i^{(2)}\right).$$

We can confirm that

$$(r_2^{\mathrm{conf}} + r_3^{\mathrm{conf}})I_{\frac{r_2^{\mathrm{conf}}}{r_2^{\mathrm{conf}}+r_3^{\mathrm{conf}}}}(P_2^{(2)}, P_3^{(2)}) = \Theta(1) \times \Theta(\epsilon^2),$$

and
$$(r_2^{\mathrm{conf}} + r_1^{\mathrm{conf}}) \geq r_2^{\mathrm{conf}} = \Theta(1),$$
which, combined with Eq.(10), implies that
$$I_{\frac{r_2^{\mathrm{conf}}}{r_2^{\mathrm{conf}} + r_1^{\mathrm{conf}}}}(P_2^{(2)}, P_1^{(2)}) = \Theta(\epsilon^2),$$
which implies $r_1^{\mathrm{conf}} = \Theta(\epsilon^2)$. $\qquad\square$

## D    Extension to wider models

In the main body of the paper, we assumed that $P \in \mathcal{P}$ and $Q \in \mathcal{Q}$ are Bernoulli or Gaussian distributions. Many parts of the results of the paper can be extended to exponential families or distributions over a support set $\mathcal{S} \subset \mathbb{R}$.

Let us consider an exponential family of form
$$\mathrm{d}P(x|\theta) = \exp(\theta^\top T(x) - A(\theta))\,\mathrm{d}F(x),$$
where $F$ is a base measure and $\theta \in \Theta \subset \mathbb{R}^d$ is a natural parameter. We assume that $A'(\theta) = \mathbb{E}_{X \sim F(\cdot|\theta)}[T(X)]$ has the inverse $(A')^{-1} : \mathrm{im}(T) \to \Theta$, where $\mathrm{im}(T)$ is the image of $T$.

Let $\mathcal{P}$ be a class of reward distributions. $\mathcal{P}$ can be the family of distributions over a known support $\mathcal{S} \subset \mathbb{R}$. We can also consider the case where $\mathcal{P}$ is the above exponential family with a possibly restricted parameter set $\Theta' \subset \Theta$. For example, $\mathcal{P}$ can be the set of Gaussian distributions with mean parameters in $[0, 1]$ and variances in $(0, \infty)$.

When we derive the lower bounds and construct algorithms, we introduce $\mathcal{Q}$ as a class of distributions corresponding to the estimated reward distributions of the arms. We set $\mathcal{Q} = \mathcal{P}$ when $\mathcal{P}$ is a family of distributions over a known support $\mathcal{S} \subset \mathbb{R}$. When we consider a natural exponential family with parameter set $\Theta' \subset \Theta$, we set $\mathcal{Q}$ as this exponential family with parameter set $\Theta$, so that the estimator of $P_i$ is always within $\mathcal{Q}$. For example, if we consider $\mathcal{P}$ as a class of Gaussians with means in $[0, 1]$ and variances in $(0, \infty)$, $\mathcal{Q}$ is the class of all Gaussians with means in $(-\infty, \infty)$ and variances in $(0, \infty)$.

In Algorithm 2, we use a convex combination of distributions $Q$ and $Q'$. The key property used in the analysis is the convexity of KL divergence between distributions. When we consider the family $\mathcal{P}$ of distributions over support set $\mathcal{S}$, the convexity
$$D(\alpha Q + (1 - \alpha)Q' \| P) \leq \alpha D(Q\|P) + (1 - \alpha)D(Q'\|P)$$
holds for any $P, Q, Q' \in \mathcal{Q}$ when we define $\alpha Q + (1 - \alpha)Q'$ as the mixture of $Q$ and $Q'$ with weight $(\alpha, 1 - \alpha)$. When $\mathcal{P}$ is the exponential family, the convexity of the KL divergence holds when $\alpha Q + (1 - \alpha)Q'$ is defined as the distribution in this family such that the expectation of the sufficient statistics $T(X)$ is equal to $\alpha \mathbb{E}_{X \sim Q}[T(X)] + (1 - \alpha)\mathbb{E}_{X \sim Q'}[T(X)]$. Note that this corresponds to taking the convex combination of the empirical means when we consider Bernoulli distributions or Gaussian distributions with a known variance.

By the convexity of the KL divergence, most parts of the analysis apply to $\mathcal{P}$ in this section and we straightforwardly obtain the following result.

**Proposition 10.** Theorems 1 and 2, Corollary 3, and Lemma 4 hold under the models $\mathcal{P}$ with the definition of the convex combination in this section.

The only part where the analysis is limited to Bernoulli or Gaussian is Theorem 5 on the PoE upper bound of the DOT algorithm. The subsequent results immediately follow if Theorem 5 is extended to the models in this section. Since the key property of the DOT algorithm in Lemma 4 on the trackability of the empirical divergence is still valid for these models, we expect that Theorem 5 can also be extended though it remains as an open question.

## E    Computational resources

We used a modern laptop (Macbook Pro) for learning $\boldsymbol{\theta}$. It took less than one hour to learn $\boldsymbol{\theta}$. For conducting a large number of simulations (i.e., Run TNN and existing algorithms for

$10^5$ times), we used a 2-CPU Xeon server of sixteen cores. It took less than twelve hours to complete simulations. We did not use a GPU for computation.

## F Implementation details

To speed up computation, the same $\boldsymbol{Q}$ was used for each $\boldsymbol{P}$ with the same optimal arm $i^*(\boldsymbol{P})$ in the mini-batches.

The final model $\boldsymbol{\theta}$ of the neural network is chosen as follows. We stored sequence of models $\boldsymbol{\theta}^{(1)}, \boldsymbol{\theta}^{(2)}, \ldots$ during training (Algorithm 3). Among these models, we chose the one with the maximum objective function $\arg\max_l \min_{(\boldsymbol{P},\boldsymbol{Q}) \in (\mathcal{P}^{\mathrm{emp}}, \mathcal{Q}^{\mathrm{emp}})} E(\boldsymbol{P}, \boldsymbol{Q}; \boldsymbol{\theta}^{(l)})$. Here, the minimum is taken over a finite dataset of size $|\mathcal{P}^{\mathrm{emp}}| = 32$ and $|\mathcal{Q}^{\mathrm{emp}}| = 10^5$.

The black lines in Figure 1 (a)–(c) representing $\exp(-t\inf_{\boldsymbol{Q}} \sum_i r_{\boldsymbol{\theta},i}(\boldsymbol{Q})D(Q_i\|P_i))$ are computed by the grid search of $\boldsymbol{Q}$ with each $Q_i$ separated by intervals of $5.0 \times 10^{-3}$.

## G Proofs

### G.1 Proofs of Theorems 1

In this section, we prove Theorem 1. This theorem as well as its proof is a special case of Theorem 2, but we solely prove Theorem 1 here since it is easier to follow.

In this proof, we write candidates of the true distributions and empirical distributions by $\boldsymbol{P} = (P_1, P_2, \ldots, P_K)$ and $\boldsymbol{Q} = (Q_1, Q_2, \ldots, Q_K)$, respectively. In this Sections G.1 and G.2, we write $\boldsymbol{P}[\mathcal{A}]$ and $\boldsymbol{Q}[\mathcal{A}]$ to denote the probability of the event $\mathcal{A}$ when the reward of each arm $i$ follows $P_i$ and $Q_i$, respectively. The entire history of the drawn arms and observed rewards is denoted by $\mathcal{H} = ((I(1), X(1)), (I(2), X(2)), \ldots, (I(T), X(T)))$. We write $X_{i,n}$ to denote the reward of the $n$-th draw of arm $i$. We define $\boldsymbol{n} = (n_1, n_2, \ldots, n_K)$ and $\boldsymbol{r} = (r_1, r_2, \ldots, r_K) = \boldsymbol{n}/T$ as the numbers of draws of $K$ arms and their fractions, respectively, for which we write $\boldsymbol{n}(\mathcal{H})$ and $\boldsymbol{r}(\mathcal{H})$ when we emphasize the dependence on the history $\mathcal{H}$.

We adopt the formulation of random rewards such that every $X_{i,m}$, the $m$-th reward of arm $i$ is randomly generated before the game begins, and if an arm is drawn, then this reward is revealed to the player. Then $X_{i,m}$ is well defined even if the arm $i$ is not drawn $m$ times.

Fix an arbitrary $\epsilon > 0$. We define sets of "typical" rewards under $\boldsymbol{Q}$: we write $\mathcal{T}_\epsilon(\boldsymbol{Q})$ to denote the event such that the rewards (some of which might not be revealed as noted above) satisfy

$$\sum_{i=1}^{K} \left| \left( n_i D(Q_i\|P_i) - \sum_{m=1}^{n_i} \log \frac{\mathrm{d}Q_i}{\mathrm{d}P_i}(X_{i,m}) \right) \right| \leq \epsilon T. \tag{11}$$

By the strong law of large numbers, $\lim_{T\to\infty} \boldsymbol{Q}[\mathcal{T}_\epsilon(\boldsymbol{Q})] = 1$.

Let $\mathcal{R}_T \subset \Delta_K$ be the set of all possible $\boldsymbol{r} = \boldsymbol{n}/T$. Since $n_i \in \{0, 1, \ldots, T\}$ we have

$$|\mathcal{R}_T| \leq (T+1)^K,$$

which is polynomial in $T$.

Consider an arbitrary algorithm $\pi$ and define the "typical" allocation $\boldsymbol{r}(\boldsymbol{Q}; \pi, \epsilon)$ and decision $J(\boldsymbol{Q}; \pi, \epsilon)$ of the algorithm for distributions $\boldsymbol{Q}$ as

$$\boldsymbol{r}(\boldsymbol{Q}; \pi, \epsilon) = \arg\max_{\boldsymbol{r} \in \mathcal{R}_T} \boldsymbol{Q}\left[\boldsymbol{r}(\mathcal{H}) = \boldsymbol{r} \middle| \mathcal{T}_\epsilon(\boldsymbol{Q})\right],$$

$$J(\boldsymbol{Q}; \pi, \epsilon) = \arg\max_{i \in [K]} \boldsymbol{Q}\left[J(T) = i \middle| \boldsymbol{r}(\mathcal{H}) = \boldsymbol{r}(\boldsymbol{Q}; \pi, \epsilon), \mathcal{T}_\epsilon(\boldsymbol{Q})\right].$$

Then we have

$$\boldsymbol{Q}\left[\boldsymbol{r}(\mathcal{H}) = \boldsymbol{r}(\boldsymbol{Q}; \pi, \epsilon) \middle| \mathcal{T}_\epsilon(\boldsymbol{Q})\right] \geq \frac{1}{|\mathcal{R}_T|}, \tag{12}$$

$$\boldsymbol{Q}\left[J(T) = J(\boldsymbol{Q}; \pi, \epsilon) \middle| \boldsymbol{r}(\mathcal{H}) = \boldsymbol{r}(\boldsymbol{Q}; \pi, \epsilon), \mathcal{T}_\epsilon(\boldsymbol{Q})\right] \geq \frac{1}{K}. \tag{13}$$

**Lemma 11.** Let $\epsilon > 0$ and algorithm $\pi$ be arbitrary. Then, for any $\boldsymbol{P}, \boldsymbol{Q}$ such that $J(\boldsymbol{Q}; \pi, \epsilon) \neq \mathcal{I}^*(\boldsymbol{P})$ it holds that

$$\frac{1}{T} \log \boldsymbol{P}[J(T) \notin \mathcal{I}^*(\boldsymbol{P})] \geq -\sum_{i=1}^{K} r_i(\boldsymbol{Q}; \pi, \epsilon) D(Q_i \| P_i) - \epsilon - \delta_{\boldsymbol{P}, \boldsymbol{Q}, \epsilon}(T)$$

for a function $\delta_{\boldsymbol{P}, \boldsymbol{Q}, \epsilon}(T)$ satisfying $\lim_{T \to \infty} \delta_{\boldsymbol{P}, \boldsymbol{Q}, \epsilon}(T) = 0$.

*Proof.* For arbitrary $\boldsymbol{Q}$ we obtain by a standard argument of a change of measures that

$$\boldsymbol{P}[J(T) \notin \mathcal{I}^*(\boldsymbol{P})]$$
$$\geq \boldsymbol{P}[\mathcal{T}_\epsilon(\boldsymbol{Q}), \boldsymbol{r}(\mathcal{H}) = \boldsymbol{r}(\boldsymbol{Q}; \pi, \epsilon), J(T) = J(\boldsymbol{Q}; \pi, \epsilon)]$$
$$= \boldsymbol{P}[\mathcal{T}_\epsilon(\boldsymbol{Q}), \boldsymbol{r}(\mathcal{H}) = \boldsymbol{r}(\boldsymbol{Q}; \pi, \epsilon)] \boldsymbol{P}[J(T) = J(\boldsymbol{Q}; \pi, \epsilon) \mid \mathcal{T}_\epsilon(\boldsymbol{Q}), \boldsymbol{r}(\mathcal{H}) = \boldsymbol{r}(\boldsymbol{Q}; \pi, \epsilon)]$$
$$= \boldsymbol{P}[\mathcal{T}_\epsilon(\boldsymbol{Q}), \boldsymbol{r}(\mathcal{H}) = \boldsymbol{r}(\boldsymbol{Q}; \pi, \epsilon)] \boldsymbol{Q}[J(T) = J(\boldsymbol{Q}; \pi, \epsilon) \mid \mathcal{T}_\epsilon(\boldsymbol{Q}), \boldsymbol{r}(\mathcal{H}) = \boldsymbol{r}(\boldsymbol{Q}; \pi, \epsilon)] \tag{14}$$
$$\geq \frac{1}{K} \boldsymbol{P}[\mathcal{T}_\epsilon(\boldsymbol{Q}), \boldsymbol{r}(\mathcal{H}) = \boldsymbol{r}(\boldsymbol{Q}; \pi, \epsilon)] \qquad \text{(by (13))}$$
$$= \frac{1}{K} \mathbb{E}_{\boldsymbol{P}} \left[ \mathbf{1}[\mathcal{H} \in \mathcal{T}_\epsilon(\boldsymbol{Q}), \boldsymbol{r}(\mathcal{H}) = \boldsymbol{r}(\boldsymbol{Q}; \pi, \epsilon)] \right]$$
$$= \frac{1}{K} \mathbb{E}_{\boldsymbol{Q}} \left[ \mathbf{1}[\mathcal{T}_\epsilon(\boldsymbol{Q}), \boldsymbol{r}(\mathcal{H}) = \boldsymbol{r}(\boldsymbol{Q}; \pi, \epsilon)] \prod_{t=1}^{T} \frac{\mathrm{d}P_{I(t)}}{\mathrm{d}Q_{I(t)}}(X(t)) \right]$$
$$\geq \frac{1}{K} \mathbb{E}_{\boldsymbol{Q}} \left[ \mathbf{1}[\mathcal{H} \in \mathcal{T}_\epsilon(\boldsymbol{Q}), \boldsymbol{r}(\mathcal{H}) = \boldsymbol{r}(\boldsymbol{Q}; \pi, \epsilon)] \right] \exp \left( -T \sum_{i=1}^{K} r_{b,i}(\boldsymbol{Q}; \pi, \epsilon) D(Q_i \| P_i) - \epsilon T \right)$$
$$\text{(by (11))}$$
$$= \frac{1}{K} \boldsymbol{Q}[\mathcal{T}_\epsilon(\boldsymbol{Q}), \boldsymbol{r}(\mathcal{H}) = \boldsymbol{r}(\boldsymbol{Q}; \pi, \epsilon)] \exp \left( -T \sum_{i=1}^{K} r_i(\boldsymbol{Q}; \pi, \epsilon) D(Q_i \| P_i) - \epsilon T \right)$$
$$\geq \frac{\boldsymbol{Q}[\mathcal{T}_\epsilon(\boldsymbol{Q})]}{K |\mathcal{R}_T|} \exp \left( -T \sum_{i=1}^{K} r_i(\boldsymbol{Q}; \pi, \epsilon) D(Q_i \| P_i) - \epsilon T \right), \qquad \text{(by (12))}$$

where (14) holds since $J(T)$ does not depend on the true distribution $\boldsymbol{P}$ given the history $\mathcal{H}$. The proof is completed by letting $\delta_{\boldsymbol{P}, \boldsymbol{Q}, \epsilon} = \log \frac{\boldsymbol{Q}[\mathcal{H} \in \mathcal{T}_\epsilon(\boldsymbol{Q})]}{K |\mathcal{R}_T|}$. $\qquad \square$

*Proof of Theorem 1.* For each $\boldsymbol{Q}$, let $\boldsymbol{r}(\boldsymbol{Q}; \{\pi_T\}, \epsilon)$, $J(\boldsymbol{Q}; \{\pi_T\}, \epsilon)$ be such that there exists a subsequence $\{T_n\}_n \subset \mathbb{N}$ satisfying

$$\lim_{n \to \infty} \boldsymbol{r}(\boldsymbol{Q}; \pi_{T_n}, \epsilon) = \boldsymbol{r}(\boldsymbol{Q}; \{\pi_T\}, \epsilon),$$
$$J(\boldsymbol{Q}; \pi_{T_n}, \epsilon) = J(\boldsymbol{Q}; \{\pi_T\}, \epsilon), \quad \forall n.$$

Such $\boldsymbol{r}(\boldsymbol{Q}; \{\pi_T\}, \epsilon) \in \Delta_K$ and $J(\boldsymbol{Q}; \{\pi_T\}, \epsilon) \in [K]$ exist since $\Delta_K$ and $[K]$ are compact. By Lemma 11, for any $J(\boldsymbol{Q}; \{\pi_T\}, \epsilon) \notin \mathcal{I}^*(\boldsymbol{P})$ we have

$$\liminf_{T \to \infty} \frac{1}{T} \log 1/\boldsymbol{P}[J(T) \notin \mathcal{I}^*(\boldsymbol{P})] \leq \liminf_{n \to \infty} \frac{1}{T_n} \log 1/\boldsymbol{P}[J(T_n) \notin \mathcal{I}^*(\boldsymbol{P})]$$
$$\leq \sum_{i=1}^{K} r_i(\boldsymbol{Q}; \{\pi_T\}, \epsilon) D(Q_i \| P_i) + \epsilon. \tag{15}$$

By taking the worst case we have

$$R(\{\pi_T\}) = \inf_{\boldsymbol{P}} H(\boldsymbol{P}) \liminf_{T \to \infty} \frac{1}{T} \log 1/\boldsymbol{P}[J(T) \notin \mathcal{I}^*(\boldsymbol{P})]$$
$$\leq \inf_{\boldsymbol{P} \in \mathcal{P}^K, \boldsymbol{Q} \in \mathcal{Q}^K : J(\boldsymbol{Q}; \{\pi_T\}, \epsilon) \notin \mathcal{I}^*(\boldsymbol{P})} H(\boldsymbol{P}) \sum_{i=1}^{K} r_i(\boldsymbol{Q}; \{\pi_T\}, \epsilon) D(Q_i \| P_i) + \epsilon.$$

By optimizing $\{\pi^T\}$ we have

$$R(\{\pi_T\}) \leq \sup_{\{\pi_T\}} \inf_{\boldsymbol{P} \in \mathcal{P}^K} H(\boldsymbol{P}) \liminf_{T \to \infty} \frac{1}{T} \log 1/\boldsymbol{P}[J(T) \notin \mathcal{I}^*(\boldsymbol{P})]$$

$$= \sup_{\boldsymbol{r}(\cdot), J(\cdot)} \sup_{\{\pi_T\}:\boldsymbol{r}(\cdot;\{\pi_T\},\epsilon)=\boldsymbol{r}(\cdot)} \inf_{\boldsymbol{P} \in \mathcal{P}^K} H(\boldsymbol{P}) \liminf_{T \to \infty} \frac{1}{T} \log 1/\boldsymbol{P}[J(T) \notin \mathcal{I}^*(\boldsymbol{P})]$$

$$\leq \sup_{\boldsymbol{r}(\cdot), J(\cdot)} \sup_{\{\pi_T\}:\boldsymbol{r}(\cdot;\{\pi_T\},\epsilon)=\boldsymbol{r}(\cdot)} \inf_{\boldsymbol{P} \in \mathcal{P}^K, \boldsymbol{Q} \in \mathcal{Q}^K:J(\boldsymbol{Q})\notin\mathcal{I}^*(\boldsymbol{P})} H(\boldsymbol{P}) \sum_{i=1}^{K} r_i(\boldsymbol{Q})D(Q_i\|P_i) + \epsilon$$

$$\text{(by (15))}$$

$$\leq \sup_{\boldsymbol{r}(\cdot), J(\cdot)} \inf_{\boldsymbol{P} \in \mathcal{P}^K, \boldsymbol{Q} \in \mathcal{Q}^K:J(\boldsymbol{Q})\notin\mathcal{I}^*(\boldsymbol{P})} H(\boldsymbol{P}) \sum_{i=1}^{K} r_i(\boldsymbol{Q})D(Q_i\|P_i) + \epsilon.$$

We obtain the desired result since $\epsilon > 0$ is arbitrary. $\qquad\square$

### G.2  Proof of Theorem 2

Theorem 2 is a generalization of Theorem 1, and we consider different candidates of empirical distributions depending on the batch.

As in the case of the proof of Theorem 1, we write $\boldsymbol{P} = (P_1, P_2, \ldots, P_i)$ and $\boldsymbol{P}[A]$ to denote a candidate of the true distributions and the probability of the event under $\boldsymbol{P}$. We divide $T$ rounds into $B$ batches, and the $b$-th batch corresponds to $(t_b, t_b + 1, \ldots, t_{b+1} - 1)$-th rounds for $b \in [B]$ and $t_b = \lfloor (b-1)T/B \rfloor + 1$. The entire history of the drawn arms and observed rewards is denoted by $\mathcal{H} = ((I(1), X(1)), (I(2), X(2)), \ldots, (I(T), X(T)))$. We write $X_{b,i,n}$ to denote the reward of the $n$-th draw of arm $i$ in the $b$-th batch. We define $\boldsymbol{n}_b = (n_{b,1}, n_{b,2}, \ldots, n_{b,K})$ and $\boldsymbol{r} = (r_{b,1}, r_{b,2}, \ldots, r_{b,K}) = \boldsymbol{n}_b/T$ as the numbers of draws of $K$ arms and their fractions in the $b$-th batch, respectively, for which we write $\boldsymbol{n}_b(\mathcal{H})$ and $\boldsymbol{r}_b(\mathcal{H})$ when we emphasize the dependence on the history $\mathcal{H}$.

We adopt the formulation of the random rewards such that every $X_{b,i,m}$, the $m$-th reward of arm $i$ in the $b$-th batch, is randomly generated before the game begins, and if an arm is drawn then this reward is revealed to the player. Then $X_{b,i,m}$ is well-defined even if arm $i$ is not drawn $m$ times in the $b$-th batch.

Fix an arbitrary $\epsilon > 0$. We define sets of "typical" rewards under $\boldsymbol{Q}^B$: we write $\mathcal{T}_\epsilon(\boldsymbol{Q}^B)$ to denote the event such that the rewards (a part of which might be unrevealed as noted above) satisfy

$$\sum_{i=1}^{K} \left| \left( n_{b,i}D(Q_{b,i}\|P_i) - \sum_{m=1}^{n_{b,i}} \log \frac{\mathrm{d}Q_{b,i}}{\mathrm{d}P_i}(X_{b,i,m}) \right) \right| \leq \epsilon T/B \qquad (16)$$

for any $b \in [B]$. By the strong law of large numbers, $\lim_{T \to \infty} \boldsymbol{Q}^B[\mathcal{T}_\epsilon^B(\boldsymbol{Q}^B)] = 1$, where $\boldsymbol{Q}^B[\cdot]$ denotes the probability under which $X_k(t)$ follows distribution $Q_{b,i}$ for $t \in \{t_b, t_b + 1, \ldots, t_{b+1} - 1\}$.

Let $\mathcal{R}_{T,B} \subset (\Delta_K)^B$ be the set of all possible $\boldsymbol{r}^B(\mathcal{H})$. Since $n_{b,i} \in \{0, 1, \ldots, t_{b+1} - t_b\}$ and $t_{b+1} - t_b \leq T/B + 1$, we see that

$$|\mathcal{R}_{T,B}| \leq (T/B + 2)^{KB},$$

which is polynomial in $T$.

Consider an arbitrary algorithm $\pi$ and define the "typical" allocation $\boldsymbol{r}^b(\boldsymbol{Q}^b; \pi, \epsilon)$ and decision $J(\boldsymbol{Q}^B; \pi, \epsilon)$ of the algorithm for distributions $\boldsymbol{Q}^b = (\boldsymbol{Q}_1, \boldsymbol{Q}_2, \ldots, \boldsymbol{Q}_b)$ as

$$\boldsymbol{r}_1(\boldsymbol{Q}^1; \pi, \epsilon) = \operatorname*{arg\,max}_{\boldsymbol{r} \in \mathcal{R}_{T,1}} \boldsymbol{Q}^1 \left[ \boldsymbol{r}_1(\mathcal{H}) = \boldsymbol{r} \big| \mathcal{T}_\epsilon(\boldsymbol{Q}^B) \right],$$

$$\boldsymbol{r}_b(\boldsymbol{Q}^b; \pi, \epsilon) = \operatorname*{arg\,max}_{\boldsymbol{r} \in \mathcal{R}_{T,b}} \boldsymbol{Q}^b \left[ \boldsymbol{r}_b(\mathcal{H}) = \boldsymbol{r} \big| \boldsymbol{r}^{b-1}(\mathcal{H}^{b-1}) = \boldsymbol{r}^{b-1}(\boldsymbol{Q}^{b-1}; \pi, \epsilon), \mathcal{T}_\epsilon(\boldsymbol{Q}^B) \right],$$

$$b = 2, 3, \ldots, B,$$

$$J(\boldsymbol{Q}^B; \pi, \epsilon) = \underset{i \in [K]}{\arg \max} \, \boldsymbol{Q}^B \left[ J(T) = i \middle| \boldsymbol{r}^B(\mathcal{H}) = \boldsymbol{r}^B(\boldsymbol{Q}^B; \pi, \epsilon), \, \mathcal{T}_\epsilon(\boldsymbol{Q}^B) \right].$$

Then we have

$$\boldsymbol{Q}^B \left[ \boldsymbol{r}^B(\mathcal{H}) = \boldsymbol{r}^B(\boldsymbol{Q}^B; \pi, \epsilon) \middle| \mathcal{T}_\epsilon(\boldsymbol{Q}^B) \right] \geq \frac{1}{|\mathcal{R}_{T,B}|}, \tag{17}$$

$$\boldsymbol{Q}^B \left[ J(T) = J(\boldsymbol{Q}^B; \pi, \epsilon) \middle| \boldsymbol{r}^B(\mathcal{H}) = \boldsymbol{r}^B(\boldsymbol{Q}^B; \pi, \epsilon), \, \mathcal{T}_\epsilon(\boldsymbol{Q}^B) \right] \geq \frac{1}{K}. \tag{18}$$

**Lemma 12.** Let $\epsilon > 0$ and algorithm $\pi$ be arbitrary. Then, for any $\boldsymbol{P}, \boldsymbol{Q}^B$ such that $J(\boldsymbol{Q}^B; \pi, \epsilon) \neq \mathcal{I}^*(\boldsymbol{P})$ it holds that

$$\frac{1}{T} \log \boldsymbol{P}[J(T) \notin \mathcal{I}^*(\boldsymbol{P})] \geq -\frac{1}{B} \sum_{b=1}^{B} \sum_{i=1}^{K} r_{b,i}(\boldsymbol{Q}^b; \pi, \epsilon) D(Q_{b,i} \| P_i) - \epsilon - \delta_{\boldsymbol{P}, \boldsymbol{Q}^B, \epsilon}(T)$$

for a function $\delta_{\boldsymbol{P}, \boldsymbol{Q}^B, \epsilon}(T)$ satisfying $\lim_{T \to \infty} \delta_{\boldsymbol{P}, \boldsymbol{Q}^B, \epsilon}(T) = 0$.

*Proof.* For arbitrary $\boldsymbol{Q}^B$ we obtain by a standard argument of a change of measures that

$$\begin{aligned}
&\boldsymbol{P}[J(T) \notin \mathcal{I}^*(\boldsymbol{P})] \\
&\geq \boldsymbol{P}[\mathcal{T}_\epsilon(\boldsymbol{Q}^B), \, \boldsymbol{r}^B(\mathcal{H}) = \boldsymbol{r}^B(\boldsymbol{Q}^B; \pi, \epsilon), \, J(T) = J(\boldsymbol{Q}^B; \pi, \epsilon)] \\
&= \boldsymbol{P}[\mathcal{T}_\epsilon(\boldsymbol{Q}^B), \, \boldsymbol{r}^B(\mathcal{H}) = \boldsymbol{r}^B(\boldsymbol{Q}^B; \pi, \epsilon)] \\
&\qquad \times \boldsymbol{P}[J(T) = J(\boldsymbol{Q}^B; \pi, \epsilon) \mid \mathcal{T}_\epsilon(\boldsymbol{Q}^B), \, \boldsymbol{r}^B(\mathcal{H}) = \boldsymbol{r}^B(\boldsymbol{Q}^B; \pi, \epsilon)] \\
&= \boldsymbol{P}[\mathcal{T}_\epsilon(\boldsymbol{Q}^B), \, \boldsymbol{r}^B(\mathcal{H}) = \boldsymbol{r}^B(\boldsymbol{Q}^B; \pi, \epsilon)] \\
&\qquad \times \boldsymbol{Q}^B[J(T) = J(\boldsymbol{Q}^B; \pi, \epsilon) \mid \mathcal{T}_\epsilon(\boldsymbol{Q}^B), \, \boldsymbol{r}^B(\mathcal{H}) = \boldsymbol{r}^B(\boldsymbol{Q}^B; \pi, \epsilon)] \tag{19} \\
&\geq \frac{1}{K} \boldsymbol{P}[\mathcal{T}_\epsilon(\boldsymbol{Q}^B), \, \boldsymbol{r}^B(\mathcal{H}) = \boldsymbol{r}^B(\boldsymbol{Q}^B; \pi, \epsilon)] \qquad\qquad \text{(by (18))} \\
&= \frac{1}{K} \mathbb{E}_{\boldsymbol{P}} \left[ \mathbf{1}[\mathcal{H} \in \mathcal{T}_\epsilon(\boldsymbol{Q}^B), \, \boldsymbol{r}^B(\mathcal{H}) = \boldsymbol{r}^B(\boldsymbol{Q}^B; \pi, \epsilon)] \right] \\
&= \frac{1}{K} \mathbb{E}_{\boldsymbol{Q}^B} \left[ \mathbf{1}[\mathcal{T}_\epsilon(\boldsymbol{Q}^B), \, \boldsymbol{r}^B(\mathcal{H}) = \boldsymbol{r}^B(\boldsymbol{Q}^B; \pi, \epsilon)] \prod_{b=1}^{B} \prod_{t=t_b}^{t_{b+1}-1} \frac{dP_{I(t)}}{dQ_{b,I(t)}}(X(t)) \right] \\
&\geq \frac{1}{K} \mathbb{E}_{\boldsymbol{Q}^B} \left[ \mathbf{1}[\mathcal{H} \in \mathcal{T}_\epsilon(\boldsymbol{Q}^B), \, \boldsymbol{r}^B(\mathcal{H}^B) = \boldsymbol{r}^B(\boldsymbol{Q}^B; \pi, \epsilon)] \right] \\
&\qquad \times \exp \left( -\frac{T}{B} \sum_{b=1}^{B} \sum_{i=1}^{K} r_{b,i}(\boldsymbol{Q}^b; \pi, \epsilon) D(Q_{b,i} \| P_i) - \epsilon T \right) \qquad \text{(by (16))} \\
&= \frac{1}{K} \boldsymbol{Q}^B \left[ \mathcal{T}_\epsilon(\boldsymbol{Q}^B), \, \boldsymbol{r}^B(\mathcal{H}^B) = \boldsymbol{r}^B(\boldsymbol{Q}^B; \pi, \epsilon) \right] \\
&\qquad \times \exp \left( -\frac{T}{B} \sum_{b=1}^{B} \sum_{i=1}^{K} r_{b,i}(\boldsymbol{Q}^b; \pi, \epsilon) D(Q_{b,i} \| P_i) - \epsilon T \right) \\
&\geq \frac{\boldsymbol{Q}^B[\mathcal{T}_\epsilon(\boldsymbol{Q}^B)]}{K|\mathcal{R}_{T,B}|} \exp \left( -\frac{T}{B} \sum_{b=1}^{B} \sum_{i=1}^{K} r_{b,i}(\boldsymbol{Q}^b; \pi, \epsilon) D(Q_{b,i} \| P_i) - \epsilon T \right), \qquad \text{(by (17))}
\end{aligned}$$

where (19) holds since $J(T)$ does not depend on the true distribution $\boldsymbol{P}$ given the history $\mathcal{H}$. The proof is completed by letting $\delta_{\boldsymbol{P}, \boldsymbol{Q}^B, \epsilon} = \log \frac{\boldsymbol{Q}^B[\mathcal{T}_\epsilon(\boldsymbol{Q}^B)]}{K|\mathcal{R}_{T,B}|}$. $\qquad \square$

*Proof of Theorem 2.* For each $\boldsymbol{Q}^B$, let $\boldsymbol{r}^B(\boldsymbol{Q}^B; \{\pi_T\}, \epsilon)$, $J(\boldsymbol{Q}^B; \{\pi_T\}, \epsilon)$ be such that there exists a subsequence $\{T_n\}_n \subset \mathbb{N}$ satisfying

$$\lim_{n \to \infty} \boldsymbol{r}^B(\boldsymbol{Q}^B; \pi_{T_n}, \epsilon) = \boldsymbol{r}^B(\boldsymbol{Q}^B; \{\pi_T\}, \epsilon),$$

$$J(\boldsymbol{Q}^B; \pi_{T_n}, \epsilon) = J(\boldsymbol{Q}^B; \{\pi_T\}, \epsilon), \quad \forall n.$$

Such $\boldsymbol{r}^B(\boldsymbol{Q}^B; \{\pi_T\}, \epsilon) \in (\Delta_K)^B$ and $J(\boldsymbol{Q}^B; \{\pi_T\}, \epsilon) \in [K]$ exist since $(\Delta_K)^B$ and $[K]$ are compact. By Lemma 12, for any $J(\boldsymbol{Q}^B; \{\pi_T\}, \epsilon) \notin \mathcal{I}^*(\boldsymbol{P})$ we have

$$\liminf_{T\to\infty} \frac{1}{T} \log 1/\boldsymbol{P}[J(T) \notin \mathcal{I}^*(\boldsymbol{P})] \leq \liminf_{n\to\infty} \frac{1}{T_n} \log 1/\boldsymbol{P}[J(T_n) \notin \mathcal{I}^*(\boldsymbol{P})]$$

$$\leq \frac{1}{B} \sum_{b=1}^{B} \sum_{i=1}^{K} r_{b,i}(\boldsymbol{Q}^b; \{\pi_T\}, \epsilon) D(Q_{b,i}\|P_i) + \epsilon. \quad (20)$$

By taking the worst case we have

$$R(\{\pi_T\}) = \inf_{\boldsymbol{P}} H(\boldsymbol{P}) \liminf_{T\to\infty} \frac{1}{T} \log 1/\boldsymbol{P}[J(T) \notin \mathcal{I}^*(\boldsymbol{P})]$$

$$\leq \inf_{\boldsymbol{P}\in\mathcal{P}^K, \boldsymbol{Q}^B\in\mathcal{Q}^{KB}: J(\boldsymbol{Q}^B; \{\pi_T\}, \epsilon)\notin\mathcal{I}^*(\boldsymbol{P})} \frac{H(\boldsymbol{P})}{B} \sum_{b=1}^{B} \sum_{i=1}^{K} r_{b,i}(\boldsymbol{Q}^b; \{\pi_T\}, \epsilon) D(Q_{b,i}\|P_i) + \epsilon.$$

By optimizing $\{\pi^T\}$ we have

$$R(\{\pi_T\}) \leq \sup_{\{\pi_T\}} \inf_{\boldsymbol{P}\in\mathcal{P}^K} H(\boldsymbol{P}) \liminf_{T\to\infty} \frac{1}{T} \log 1/\boldsymbol{P}[J(T) \notin \mathcal{I}^*(\boldsymbol{P})]$$

$$= \sup_{\boldsymbol{r}^B(\cdot), J(\cdot)} \sup_{\{\pi_T\}: \boldsymbol{r}^B(\cdot; \{\pi_T\}, \epsilon)=\boldsymbol{r}^B(\cdot)} \inf_{\boldsymbol{P}\in\mathcal{P}^K} \frac{H(\boldsymbol{P})}{B} \liminf_{T\to\infty} \frac{1}{T} \log 1/\boldsymbol{P}[J(T) \notin \mathcal{I}^*(\boldsymbol{P})]$$

$$\leq \sup_{\boldsymbol{r}^B(\cdot), J(\cdot)} \sup_{\{\pi_T\}: \boldsymbol{r}^B(\cdot; \{\pi_T\}, \epsilon)=\boldsymbol{r}^B(\cdot)} \inf_{\boldsymbol{P}\in\mathcal{P}^K, \boldsymbol{Q}^B\in\mathcal{Q}^{KB}: J(\boldsymbol{Q}^B)\notin\mathcal{I}^*(\boldsymbol{P})} \frac{H(\boldsymbol{P})}{B} \sum_{b=1}^{B} \sum_{i=1}^{K} r_{b,i}(\boldsymbol{Q}^b) D(Q_{b,i}\|P_i) + \epsilon$$
$$\text{(by (20))}$$

$$\leq \sup_{\boldsymbol{r}^B(\cdot), J(\cdot)} \inf_{\boldsymbol{P}\in\mathcal{P}^K, \boldsymbol{Q}^B\in\mathcal{Q}^{KB}: J(\boldsymbol{Q}^B)\notin\mathcal{I}^*(\boldsymbol{P})} \frac{H(\boldsymbol{P})}{B} \sum_{b=1}^{B} \sum_{i=1}^{K} r_{b,i}(\boldsymbol{Q}^b) D(Q_{b,i}\|P_i) + \epsilon.$$

We obtain the desired result since $\epsilon > 0$ is arbitrary. $\qquad\square$

### G.3 Proof of Corollary 3

*Proof of Corollary 3.* We have

$$R_B^{\mathrm{go}}$$

$$:= \sup_{\boldsymbol{r}^B(\boldsymbol{Q}^B), J(\boldsymbol{Q}^B)} \inf_{\boldsymbol{Q}^B} \inf_{\boldsymbol{P}: J(\boldsymbol{Q}^B)\notin\mathcal{I}^*(\boldsymbol{P})} \frac{H(\boldsymbol{P})}{B} \sum_{i\in[K], b\in[B]} r_{b,i} D(Q_{b,i}\|P_i)$$

$$\leq \sup_{\boldsymbol{r}^B(\boldsymbol{Q}^B), J(\boldsymbol{Q}^B)} \inf_{\boldsymbol{Q}^B: \boldsymbol{Q}_1=\boldsymbol{Q}_2=\cdots=\boldsymbol{Q}_B} \inf_{\boldsymbol{P}: J(\boldsymbol{Q}^B)\notin\mathcal{I}^*(\boldsymbol{P})} \frac{H(\boldsymbol{P})}{B} \sum_{i\in[K], b\in[B]} r_{b,i} D(Q_{b,i}\|P_i) \quad \text{(inf over a subset).}$$

$$= \sup_{\boldsymbol{r}^B(\boldsymbol{Q}), J(\boldsymbol{Q})} \inf_{\boldsymbol{Q}} \inf_{\boldsymbol{P}: J(\boldsymbol{Q})\notin\mathcal{I}^*(\boldsymbol{P})} H(\boldsymbol{P}) \sum_{i\in[K]} \left( \frac{1}{B} \sum_{b\in[B]} r_{b,i} \right) D(Q_i\|P_i)$$
$$\text{(by denoting } \boldsymbol{Q} = \boldsymbol{Q}_1 = \boldsymbol{Q}_2 = \ldots \boldsymbol{Q}_B\text{)}$$

$$= \sup_{\boldsymbol{r}(\boldsymbol{Q}), J(\boldsymbol{Q})} \inf_{\boldsymbol{Q}} \inf_{\boldsymbol{P}: J(\boldsymbol{Q})\notin\mathcal{I}^*(\boldsymbol{P})} H(\boldsymbol{P}) \sum_{i\in[K]} r_i D(Q_i\|P_i)$$
$$\text{(by letting } r_i = (1/B) \sum_b r_{b,i}\text{)}$$

$$= R^{\mathrm{go}} \quad \text{(by definition).}$$

$$\square$$

### G.4 Additional lemmas

The following lemma is used to derive the regret bound.

**Lemma 13.** Assume that we run Algorithm 2. Then, for any $B_C \in K, K+1, \ldots, B$, it follows that

$$\sum_{i,b \in [B_C]} r_{b,i} D(Q_{b,i} || P_i) \geq \sum_{i,a \in [B_C - K]} r_{a,i}^* D(Q_{a,i}' || P_i) + \sum_{i \in [K]} D(Q_{B_C - K+1, i}' || P_i). \quad (21)$$

*Proof of Lemma 13.* We use induction over $B_C \geq K$. (i) It is trivial to derive Eq. (21) for $B_C = K$. (ii) Assume that Eq. (21) holds for $B_C$. In batch $B_C + 1$, the algorithm draws arms in accordance with allocation $\boldsymbol{r}_{B_C+1} = \boldsymbol{r}_{B_C-K+1}^*$. We have,

$$\sum_{i \in [K], b \in [B_C+1]} r_{b,i} D(Q_{b,i} || P_i)$$

$$\geq \sum_{i \in [K], a \in [B_C - K]} r_{a,i}^* D(Q_{a,i}' || P_i) + \sum_{i \in [K]} D(Q_{B_C - K+1, i}' || P_i) + \underbrace{\sum_i r_{B_C+1, i} D(Q_{B_C+1, i} || P_i)}_{\text{Batch } B_C + 1}$$

(by the assumption of the induction)

$$= \sum_i \left( \sum_{a \in [B_C - K]} r_{a,i}^* D(Q_{a,i}' || P_i) + r_{B_C - K+1, i}^* D(Q_{B_C - K+1, i}' || P_i) \right) + \sum_i \left( 1 - r_{B_C - K+1, i}^* \right) D(Q_{B_C - K+1, i}' || P_i)$$

$$+ \sum_i r_{B_C+1, i} D(Q_{B_C+1, i} || P_i)$$

$$= \sum_i \left( \sum_{a \in [B_C - K]} r_{a,i}^* D(Q_{a,i}' || P_i) + r_{B_C - K+1, i}^* D(Q_{B_C - K+1, i}' || P_i) \right) + \sum_i \left( 1 - r_{B_C+1, i} \right) D(Q_{B_C - K+1, i}' || P_i)$$

$$+ \sum_i r_{B_C+1, i} D(Q_{B_C+1, i} || P_i)$$

(by definition)

$$= \sum_i \left( \sum_{a \in [B_C - K]} r_{a,i}^* D(Q_{a,i}' || P_i) + r_{B_C - K+1, i}^* D(Q_{B_C - K+1, i}' || P_i) \right) + \sum_i D(Q_{B_C - K+2, i}' || P_i)$$

(by Jensen's inequality and $Q_{B_C - K+2, i}' = r_{B_C+1, i} Q_{B_C+1, i} + (1 - r_{B_C+1, i}) Q_{B_C - K+1, i}'$)

$$= \sum_i \sum_{a \in [B_C - K+1]} r_{a,i}^* D(Q_{a,i}' || P_i) + \sum_i D(Q_{B_C - K+2, i}' || P_i).$$

$\square$

### G.5 Proof of Lemma 4

*Proof of Lemma 4.*

$$\sum_{i,b \in [B+K-1]} r_{b,i} D(Q_{b,i} || P_i) \geq \sum_{i,b \in [B-1]} r_{b,i}^* D(Q_{b,i}' || P_i) + \sum_i D(Q_{B,i}' || P_i). \quad \text{(by (21))}$$

$$\geq \sum_{i,b \in [B]} r_{b,i}^* D(Q_{b,i}' || P_i)$$

$$\geq \frac{B(R_B^{\text{go}} - \epsilon)}{H(\boldsymbol{P})} \quad \text{(by definition of } \epsilon\text{-optimal solution).}$$

$\square$

### G.6 Proof of Theorem 5

*Proof of Theorem 5, Bernoulli rewards.* Since the reward is binary, the possible values that $Q_{b,i}$ lie in a finite set

$$\mathcal{V} = \left\{ \frac{l}{m} : l \in \mathbb{N}, m \in \mathbb{N}^+ \right\},$$

where it is easy to prove $|\mathcal{V}| \leq (T/(B+K-1)+2)^2 \leq (T/B+2)^2$. We have

$$\mathbb{P}[J(T) \notin \mathcal{I}^*(\boldsymbol{P})] = \sum_{\boldsymbol{V}_1,\ldots,\boldsymbol{V}_B \in \mathcal{V}^K} \mathbb{P}\left[ J(T) \notin \mathcal{I}^*(\boldsymbol{P}), \bigcap_b \{\boldsymbol{Q}_b = \boldsymbol{V}_b\} \right]$$

$$= \sum_{\boldsymbol{V}_1,\ldots,\boldsymbol{V}_B \in \mathcal{V}^K : J^*(\boldsymbol{V}_1,\ldots,\boldsymbol{V}_B) \notin \mathcal{I}^*(\boldsymbol{P})} \mathbb{P}\left[ \bigcap_b \{\boldsymbol{Q}_b = \boldsymbol{V}_b\} \right].$$

By using the Chernoff bound, we have

$$\mathbb{P}\left[ Q_{b,i} = V_{b,i} \middle| \bigcap_{b' \in [b-1]} \{\boldsymbol{Q}_{b'} = \boldsymbol{V}_{b'}\} \right] \leq e^{-\frac{T'}{B+K-1} r_{b,i} D(V_{b,i}\|P_i)}, \tag{22}$$

and thus

$$\mathbb{P}\left[ \bigcap_b \{\boldsymbol{Q}_b = \boldsymbol{V}_b\} \right]$$

$$= \prod_b \mathbb{P}\left[ \boldsymbol{Q}_b = \boldsymbol{V}_b \middle| \bigcap_{b'=1}^{b-1} \{\boldsymbol{Q}_{b'} = \boldsymbol{V}_{b'}\} \right]$$

$$\leq \prod_b e^{-\frac{T'}{B+K-1} \sum_i r_{b,i} D(V_{b,i}\|P_i)} \quad \text{(by Eq. (22))}$$

$$= e^{-\frac{T'}{B+K-1} \sum_{b,i} r_{b,i} D(V_{b,i}\|P_i)}. \tag{23}$$

Furthermore,

$$\mathbb{P}\left[ \bigcap_b \{\boldsymbol{Q}_b = \boldsymbol{V}_b\} \right]$$

$$= \mathbb{P}\left[ \bigcap_b \{\boldsymbol{Q}_b = \boldsymbol{V}_b\}, \sum_{i,b \in [B+K-1]} r_{b,i} D(Q_{b,i}\|P_i) \geq \frac{B(R_B^{\text{go}} - \epsilon)}{H(\boldsymbol{P})} \right]$$

(by Lemma 4).

$$= \mathbb{P}\left[ \bigcap_b \{\boldsymbol{Q}_b = \boldsymbol{V}_b\} \right] \mathbb{P}\left[ \sum_{i,b \in [B+K-1]} r_{b,i} D(Q_{b,i}\|P_i) \geq \frac{B(R_B^{\text{go}} - \epsilon)}{H(\boldsymbol{P})} \middle| \bigcap_b \{\boldsymbol{Q}_b = \boldsymbol{V}_b\} \right]$$

$$= \mathbb{P}\left[ \bigcap_b \{\boldsymbol{Q}_b = \boldsymbol{V}_b\} \right] \mathbb{P}\left[ \sum_{i,b \in [B+K-1]} r_{b,i} D(V_{b,i}\|P_i) \geq \frac{B(R_B^{\text{go}} - \epsilon)}{H(\boldsymbol{P})} \right]$$

$$= \mathbb{P}\left[ \bigcap_b \{\boldsymbol{Q}_b = \boldsymbol{V}_b\} \right] \mathbb{E}\left[ \mathbf{1}\left[ \sum_{i,b \in [B+K-1]} r_{b,i} D(V_{b,i}\|P_i) \geq \frac{B(R_B^{\text{go}} - \epsilon)}{H(\boldsymbol{P})} \right] \right]$$

$$\leq e^{-\frac{T'}{B+K-1}\sum_{b,i} r_{b,i}D(V_{b,i}||P_i)}\mathbb{E}\left[\mathbf{1}\left[\sum_{i,b\in[B+K-1]} r_{b,i}D(V_{b,i}||P_i) \geq \frac{B(R_B^{\text{go}}-\epsilon)}{H(\boldsymbol{P})}\right]\right]$$

(by Eq. (23))

$$= \mathbb{E}\left[e^{-\frac{T'}{B+K-1}\sum_{b,i} r_{b,i}D(V_{b,i}||P_i)}\mathbf{1}\left[\sum_{i,b\in[B+K-1]} r_{b,i}D(V_{b,i}||P_i) \geq \frac{B(R_B^{\text{go}}-\epsilon)}{H(\boldsymbol{P})}\right]\right]$$

$$\leq \mathbb{E}\left[e^{-\frac{T'}{B+K-1}\frac{B(R_B^{\text{go}}-\epsilon)}{H(\boldsymbol{P})}}\right]$$

$$= e^{-\frac{T'}{B+K-1}\frac{B(R_B^{\text{go}}-\epsilon)}{H(\boldsymbol{P})}}. \tag{24}$$

Therefore, we have

$$\mathbb{P}[J(T) \notin \mathcal{I}^*(\boldsymbol{P})]$$

$$\leq \sum_{\boldsymbol{V}_1,\ldots,\boldsymbol{V}_B \in \mathcal{V}^K} e^{-\frac{B}{B+K-1}\frac{(R_B^{\text{go}}-\epsilon)T'}{H(\boldsymbol{P})}}$$

(by Eq. (24))

$$\leq (T/B+2)^{2KB}e^{-\frac{B}{B+K-1}\frac{(R_B^{\text{go}}-\epsilon)T'}{H(\boldsymbol{P})}}.$$

Here, $\log((T/B+2)^{2KB}) = o(T)$ to $T$ when we consider $K, B$ as constants.

$\square$

*Proof of Theorem 5, Normal rewards.* For the ease of discussion, we assume unit variance $\sigma = 1$. Extending it to the case of common known variance $\sigma$ is straightforward. Let

$$\mathcal{B} = \bigcup_{i,b}\{|Q_{b,i}| \geq T\}.$$

Then, it is easy to see

$$\mathbb{P}[\mathcal{B}] = T^{2KB}O(e^{-T^2/2}),$$

which is negligible because $\log(1/\mathbb{P}[\mathcal{B}])/T$ diverges.

The PoE is bounded as

$$\mathbb{P}[J(T) \notin \mathcal{I}^*(\boldsymbol{P})] = \mathbb{P}\left[J(T) \notin \mathcal{I}^*(\boldsymbol{P}), \mathcal{B}^c\right] + \mathbb{P}[\mathcal{B}]$$

We have,

$$\mathbb{P}\left[J(T) \notin \mathcal{I}^*(\boldsymbol{P}), \mathcal{B}^c\right]$$
$$= \int_{-T}^{T}\cdots\int_{-T}^{T}\mathbf{1}[J(T) \notin \mathcal{I}^*(\boldsymbol{P})]p(\boldsymbol{Q}_B|\boldsymbol{Q}_{B-1}\ldots\boldsymbol{Q}_1)\,\mathrm{d}\boldsymbol{Q}_B\ldots p(\boldsymbol{Q}_B|\boldsymbol{Q}_{B-1}\ldots\boldsymbol{Q}_1)\,\mathrm{d}\boldsymbol{Q}_b\ldots p(\boldsymbol{Q}_1)\,\mathrm{d}\boldsymbol{Q}_1. \tag{25}$$

Here,

$$p(\boldsymbol{Q}_b|\boldsymbol{Q}_{b-1}\ldots\boldsymbol{Q}_1) = \prod_{i\in[K]}\frac{n_{b,i}}{\sqrt{2\pi}}\exp\left(-\frac{n_{b,i}(Q_{b,i}-P_i)^2}{2}\right)$$

$$= \prod_{i\in[K]}\frac{n_{b,i}}{\sqrt{2\pi}}\exp\left(-n_{b,i}D(Q_{b,i}||P_i)\right)$$

$$\leq \prod_{i\in[K]}T\exp\left(-n_{b,i}D(Q_{b,i}||P_i)\right).$$

Finally, we have

$$(25) \leq T^{BK} \int_{-T}^{T} \cdots \int_{-T}^{T} \mathbf{1}[J(T) \notin \mathcal{I}^*(\boldsymbol{P})] \prod_{i \in [K]} \prod_{b \in [B+K-1]} \exp\left(-n_{b,i} D(Q_{b,i} \| P_i)\right) \mathrm{d}\boldsymbol{Q}_B \ldots \mathrm{d}\boldsymbol{Q}_1$$

$$\leq T^{BK} \int_{-T}^{T} \cdots \int_{-T}^{T} \mathbf{1}[J(T) \notin \mathcal{I}^*(\boldsymbol{P})] \prod_{i \in [K]} \prod_{b \in [B+K-1]} \exp\left(-\frac{T' r^{(b,i)}}{B+K-1} D(Q_{b,i} \| P_i)\right) \mathrm{d}\boldsymbol{Q}_B \ldots \mathrm{d}\boldsymbol{Q}_1$$

$$\leq T^{BK} \int_{-T}^{T} \cdots \int_{-T}^{T} \mathbf{1}[J(T) \notin \mathcal{I}^*(\boldsymbol{P})] \exp\left(-\frac{B}{B+K-1} \frac{(R_B^{\mathrm{go}} - \epsilon)T'}{H(\boldsymbol{P})}\right) \mathrm{d}\boldsymbol{Q}_B \ldots \mathrm{d}\boldsymbol{Q}_1 \quad \text{(by Lemma 4)}$$

$$\leq T^{BK} \int_{-T}^{T} \cdots \int_{-T}^{T} \exp\left(-\frac{B}{B+K-1} \frac{(R_B^{\mathrm{go}} - \epsilon)T'}{H(\boldsymbol{P})}\right) \mathrm{d}\boldsymbol{Q}_B \ldots \mathrm{d}\boldsymbol{Q}_1$$

$$\leq T^{BK} (2T)^{BK} \exp\left(-\frac{B}{B+K-1} \frac{(R_B^{\mathrm{go}} - \epsilon)T'}{H(\boldsymbol{P})}\right).$$

$\square$

### G.7 Proof of Theorem 7

*Proof of Theorem 7.* We first show that the limit

$$R_\infty^{\mathrm{go}} = \lim_{B \to \infty} R_B^{\mathrm{go}}$$

exists. Namely, for any $\eta > 0$ there exists $B_0 \in \mathbb{N}$ such that for any $B_1 > B_0$ we have

$$|R_{B_0}^{\mathrm{go}} - R_{B_1}^{\mathrm{go}}| \leq \eta.$$

Theorem 5 implies that Algorithm 2 with $B = B_0$ and $\epsilon = \eta/2$ satisfies[15]

$$\liminf_{T \to \infty} \frac{\log(1/\mathbb{P}[J(T) \notin \mathcal{I}^*(\boldsymbol{P})])}{T} \geq \frac{B_0}{B_0 + K - 1} \frac{R_{B_0}^{\mathrm{go}} - \eta/2}{H(\boldsymbol{P})},$$

and thus

$$\inf H(\boldsymbol{P}) \liminf_{T \to \infty} \frac{\log(1/\mathbb{P}[J(T) \notin \mathcal{I}^*(\boldsymbol{P})])}{T} \geq \frac{B_0}{B_0 + K - 1} \left(R_{B_0}^{\mathrm{go}} - \frac{\eta}{2}\right). \qquad (26)$$

Moreover, Theorem 2 implies that any algorithm satisfies

$$\inf H(\boldsymbol{P}) \limsup_{T \to \infty} \frac{\log(1/\mathbb{P}[J(T) \notin \mathcal{I}^*(\boldsymbol{P})])}{T} \leq R_{B_1}^{\mathrm{go}}. \qquad (27)$$

Combining Eq. (26) and Eq. (27), we have

$$\frac{B_0}{B_0 + K - 1} \left(R_{B_0}^{\mathrm{go}} - \eta/2\right) \leq R_{B_1}^{\mathrm{go}}$$

and thus

$$R_{B_0}^{\mathrm{go}} \leq R_{B_1}^{\mathrm{go}} + \frac{\eta}{2} + \frac{K-1}{B_0 + K - 1} R_{B_0}^{\mathrm{go}}$$

$$\leq R_{B_1}^{\mathrm{go}} + \frac{\eta}{2} + \frac{K-1}{B_0 + K - 1} R^{\mathrm{go}} \quad \text{(by Corollary 3)}$$

$$\leq R_{B_1}^{\mathrm{go}} + \frac{\eta}{2} + \frac{\eta}{2} \quad \text{(by } K \geq 2 \text{, by taking } B_0 \geq 2KR^{\mathrm{go}}/\eta\text{)}$$

---

[15]Strictly speaking, Algorithm 2 depends on $T$, and we take sequence of the algorithm $(\pi_{\mathrm{DOT},T})_{T=1,2,\ldots}$.

$$\leq R^{\text{go}}_{B_1} + \eta.$$

By swapping $B_0, B_1$, it is easy to show that

$$R^{\text{go}}_{B_1} \leq R^{\text{go}}_{B_0} + \eta,$$

and thus

$$|R^{\text{go}}_{B_0} - R^{\text{go}}_{B_1}| \leq \eta,$$

which implies that the limit exists. It is easy to confirm that the performance of Algorithm 2 with any $B \geq 2K R^{\text{go}}/\eta$ and $\epsilon = \eta/2$ satisfies Eq. (6). $\qquad \square$