# OpenReview forum: "Minimax Optimal Algorithms for Fixed-Budget Best Arm Identification"
_NeurIPS.cc/2022/Conference — NeurIPS 2022 Accept_

### Official Review · Reviewer_tDsK · 2022-07-07

**Rating:** 7
**Confidence:** 4
**Soundness:** 3 good
**Presentation:** 4 excellent
**Contribution:** 3 good

**Summary:**

This work studies best arm identification (BAI) under fixed budget. Two rates (of lower bounds) $R^{\text{go}}$ and $R^{\text{go}}_\infty$ are derived, where a matching algorithm for $R^{\text{}go}_\infty$ is proposed despite computationally not trackable. Then a heuristic approach is proposed to empirically match $R^{\text{go}}$.

**Questions:**

See weakness.

**Limitations:**

This work is most on maths and algorithm perspective, its social impact depends on the application of the proposed approach, which goes beyond the scope of current paper.

**Strengths And Weaknesses:**

(+) This work studies lower bounds on relatively less studied BAI with fixed budget. Since there are not many works in this area, these results are useful and can guide the design of new algorithms.

(+) A lower bound matching algorithm for $R^{\text{}go}_\infty$ is proposed. Though still can be improved by reducing computationally overhead, it is still inspiring for the problem at hand.

(-) Successive halving (SH) is not included in numerical tests.

---

> ### Author Response · Authors · 2022-08-01
> **Response to Official Review of Paper7640 by Reviewer tDsK**
>
> Thank you so much for your careful reading and comments.
>
> > (-) Successive halving (SH) is not included in numerical tests.
>
> SH would be indeed the candidate to be compared for large K. Still, the current simulation is unfortunately limited to K=3 due to the difficulty of learning NN, where SH would not make much difference with successive rejection.

---

### Official Review · Reviewer_BbC9 · 2022-07-09

**Rating:** 4
**Confidence:** 4
**Soundness:** 3 good
**Presentation:** 3 good
**Contribution:** 2 fair

**Summary:**

The paper characterizes the optimal rate of error in the fixed budget best arm identification setting. It introduces two tracking algorithms as an endeavor for achieving the best error rate; the first algorithm is not practical but it theoretically achieves the optimal rate. The second algorithm solves a simpler optimization problem but it is computationally expensive.

**Questions:**

1. Considering the similarity of your results to the one in Kaufmann et al. (2016) ($\sup \inf \left(H(P)\sum r D(Q \| P) ... \right)$ is first introduced in there), how is your analysis are different than them?
2. How do your bounds compare to Carpentier and Locatelli (2016)? (See Major comment 1)

Please also see the "Major comments" section above.

**Limitations:**

Yes.

**Strengths And Weaknesses:**

**Strengths*
1. The problem considers an important open question in the BAI literature, namley FB BAI.
2. The proposed algorithms seem to achieve the oracle rate in the experiments.


**Major comments**
1. The positioning of the paper seems inadequate. Carpentier and Locatelli (2016) already discussed the optimal rate for FB BAI problems. It could improve the paper to discuss how the work is different than theirs. Line 55 seems to only mention the fact that the constants in Carpentier and Locatelli (2016) are loose. How does your bound $R^{\text{go}}_{...}$ compare to theirs as it does not have a closed form?
2. The difference between FC and FB settings is an essential motivating point for designing FB BAI algorithms as there are already several near-optimal FC BAI algorithms in the literature. However, the discussion starting at Line 32 (and Appendix D) does not convey the message and needs clarifications. For instance, what is "instant-wise optimal"? Following up, how did you get the expression after Line 422?
3. A concrete example for Remark 1 could explain what "reasonably" mean therein.
4. Remark 2, Line 167: Why does $R(\pi)$ become zero otherwise?
5. How can we let $B\to\infty$ while it seems $B$ is the integer number of batches which needs to be $B\leq T$ ?
6. I could not understand the crux of Algorithm 2. Why is this algorithm called "delayed" ?

**Minor comments**
1. Submiting the code could improve the reproducibility of the results.
2. Lin 159: It seems like $r^*(Q...$ should be used instead of $r(Q...$
3. Line 197: Do you have the analysis showing that ignoring the last rounds does not hurt the results?
4. Line 204: what does "open $n_{b,i}$"  means?
5. Line 222: Is it $B \mathbb{N}$ instead of $b\in \mathbb{N}$ ?

---

> ### Author Response · Authors · 2022-08-01
> **Response to Official Review of Paper7640 by Reviewer BbC9**
>
> We thank the reviewer for the careful reading. Below, we answer the questions as well as major/minor comments.
>
> > Considering the similarity of your results to the one in Kaufmann et al. (2016 ($\sup\inf(H(P)∑rD(Q|P)...$)  is first introduced in there), how is your analysis are different than them?
>
> The bound of a form $\sup_r \inf_{P,Q} (H(P)\sum rD(Q|P))$ did not appear in the papers by Kaufmann et al. (2016), where the fixed-confidence (FC) setting is mainly discussed. One of the most significant contributions of our paper is to show that the FB rate involves a tradeoff. Unlike FC, how an algorithm can perform better for some P depends on the performance required for another instance $P’$.
>
> > How do your bounds compare to Carpentier and Locatelli (2016)? (See Major comment 1)
>
> Carpentier and Locatelli 2016 show the dependence of $R^{go}$ for large $K$, whereas we discuss the best possible value of $R^{go}$ given $K$. Note also that for a fixed $K$, Carpentier and Locatelli 2016 is loose in terms of the constant at least in the order of $10^2$ (as discussed in Section 1.1).
>
>
> > The difference between FC and FB settings is an essential motivating point for designing FB BAI algorithms as there are already several near-optimal FC BAI algorithms in the literature. However, the discussion starting at Line 32 (and Appendix D) does not convey the message and needs clarifications. For instance, what is "instant-wise optimal"?
>
> Instance-wise optimal means an optimal algorithm for any instance $P$. We showed that, in FB, the performance of an algorithm on $P$ limits the algorithm's performance on another instance $P’$. Furthermore, we show $K=3$ instances below where the FC optimal algorithm performs poorly in FB for some instances. Let $P = (0.6, 0.5, 0.5-\epsilon)$ and $P' = (0.4, 0.5, 0.5-\epsilon)$. Let $A(\delta)$ be a parametrized algorithm. Assume that we run $T$ rounds the parameter $P'$ and this algorithm draws in accordance with the optimal rate of the fixed confidence BAI (GK 2016, (8) in our supplementary). In this case, it would only draw arm 1 for $O(\epsilon^2)$ times and recommend arm 2 with probability $1-o(1)$. A standard change of measure argument states that if we run $A(\delta)$ on the parameter $P$, with probability at least $(1-o(1))\exp(-O(\epsilon^2)T)$ it recommends arm 2 (which is incorrect under $P$), and thus the rate is only $\exp(-O(\epsilon^2)T/H_1(P))$, which can be arbitrarily worse by taking small $\epsilon$ in view of the fixed-budget BAI.
>
>
> > A concrete example for Remark 1 could explain what "reasonably" mean therein.
>
> For example, if we take $H(P) = 1/\Delta$ (rather than $\Delta^2$ of $H_1(P)$), in such a case rate $\exp(-R^{go}T/H)$ is not achievable around small gap and $R^{go} = 0$. Such a complexity $H(P)$ is not very reasonable in view of our minimax bound.
>
> > Remark 2, Line 167: Why does $R(\pi)$ become zero otherwise?
>
> If we use an algorithm such that $J(Q) \neq i^*(Q)$ for some $Q$, then $D(Q||P)=0$ holds independent of the algorithm $\pi$ when $P=Q$, which is included in the domain of $P$ in the inner infimum of (2) in this case. Then (2) immediately becomes zero.
>
>
> > How can we let $B$ goes $\infty$ while it seems $T$ is the integer number of batches which needs to be $B < T$?
>
> Essentially, $B, T$ such as $B=\sqrt{T}$ (or any $o(T)$ function that diverges) works. In particular, we derive the performance of the algorithm with finite $B, T$. (F.5 Proof of Theorem 5) then take limit $T \rightarrow \infty$. Then take $B \rightarrow \infty$ at Theorem 7.
>
> > I could not understand the crux of Algorithm 2. Why is this algorithm called "delayed"?
>
> The batch-oracle algorithm lower bound (that bounds the performance of any algorithm) is calculating the allocation $r_b$ based on the empirical average of the $Q_b$, which is not available until the end of batch $b$. Algorithm 2 spends the first $K$ batches to construct $Q’$ and uses $Q_{b-K}'$ ($=$ information up to $Q_{b-1}$, which is available before we start with batch $b$) instead of $Q_b$.
>
>
> > Do you have the analysis showing that ignoring the last rounds does not hurt the results?
>
> In deriving $R^{go}_\infty$, we make $B$ as well as $T/B$ goes to infinity, so flooring on $T/B$ does not hurt the asymptotic rate.
>
> > Line 204: what does "open" means?
>
> We wrote "open" when the samples are actually used to determine the allocation. In other words, observations do not affect the allocation until opened (though the actual procedure is slightly different as written in Line 200 for a technical reason). We will make it clear in the revised version.
>
> Thank you for many other useful comments.

---

> > ### Comment · Reviewer_BbC9 · 2022-08-03
> > **Comparison to FC and Tracking algorithms**
> >
> > A lower bound of the form similar to yours appears in Garivier and Kaufmann, (2016), as you also mention in Appendix D. Also, Kaufmann et al. (2016) consider the FB setting and derive a lower bound for it, which parse closely to your bounds.
> >
> > *"One of the most significant contributions of our paper is to show that the FB rate involves a tradeoff. Unlike FC, how an algorithm can perform better for some P depends on the performance required for another instance P' "*. Why is this the case? Where is this explained in the paper?
> >
> > Your example here is similar to your analysis in Appendix D, which needs clarification. As I mentioned in Major Comment 2, Appendix D does not convey the message and it is hard to parse. For instance, considering your example here, why does D-tracking *"would only draw arm 1 for $O(\epsilon^2)$ times"* ? Also, how do you apply the change of measure? Please explain the argument in Appendix D, especially, the expression after Eq. (9).
> >
> >
> > I suggest just using "instant optimal" instead of "instant-wise optimal".

---

> > > ### Author Response · Authors · 2022-08-04
> > > **Answer on Comparison to FC and Tracking algorithms**
> > >
> > > Thank you for the additional questions. We answer to each question.
> > >
> > > > A lower bound of the form similar to yours appears in Garivier and Kaufmann, (2016), as you also mention in Appendix D. Also, Kaufmann et al. (2016) consider the FB setting and derive a lower bound for it, which parse closely to your bounds.
> > >
> > > The following discusses the difference between the FC rate (Garivier and Kaufmann 2016 (GK2016)) and the minimax FB rate in our paper. The FC rate $C^{conf}(P)$ does not involve $H(P)$, and there are several FC algorithms (such as D-tracking) that achieve the sample complexity of $C^{conf}(P) \log(1/\delta) + o(\log(1/\delta))$ for **all** $P$. Assume that a FB algorithm has rate $R$ at parameter $P$. Eq. (2) limits such an allocation $r(\cdot)$ must satisfy
> > > $$\inf_Q H(P) \sum_i r_i(Q) D(Q_i || P_i) \ge R $$
> > > which limits the performance of the algorithm for other instances $P’$ by limiting the form of $r(\cdot)$ (= tradeoff between performance in $P$ and $P'$).
> > > If
> > > $$\inf_Q H(P) \sum_i r_i(Q) D(Q_i || P_i) \ge R$$
> > > $$\inf_Q H(P’) \sum_i r_i(Q) D(Q_i || P_i’) \ge R$$
> > > $$\inf_Q H(P’’) \sum_i r_i(Q) D(Q_i || P_i’’) \ge R$$
> > > are simultaneously achievable for instances $P, P’, P’’,…,$ by some allocation $r(\cdot)$, then rate $R$ is achievable among these instances $P, P’, P’’…$. If this rate $R$ is achievable for all $P \in \mathcal{P}$, we consider it as a minimax rate that we study in this paper.
> > >
> > > > "would only draw arm 1 for $O(\epsilon^2)$ times"? Also, how do you apply the change of measure?
> > >
> > > Let $r_i(P’)$ be an allocation that achieves (8). In this allocation, it is easy to see that $r_2, r_3 = \Theta(1)$. The following shows $r_1 = O(\epsilon^2)$.
> > > Lemma 4 in GK2016 states that, letting
> > > $$I_\alpha(P_1', P_2') = \alpha D(P_1'|| \alpha P_1' + (1-\alpha) P_2') + (1-\alpha) D(P_2' || \alpha P_1' + (1-\alpha) P_2'),$$
> > >  we have
> > > $$(r_2 + r_1) I_{r_2/(r_1+r_2)}(P_2’, P_1’) = (r_2 + r_3) I_{r_2/(r_2+r_3)}(P_2’, P_3’),$$
> > >  which, by using By using $d(P_1', P_2') = O( (P_1'-P_2')^2)$, $r_2 + r_1 = O(1)$, $I_{r_2/(r_1+r_2)}(P_2’, P_1’) = O(r_1)$, $r_2 + r_3 = O(1)$, $I_{r_2/(r_2+r_3)}(P_2’, P_3’) = O(\epsilon^2)$, implies $r_1 = O(\epsilon^2)$.
> > >
> > > The change of measure between the probabilities $\mathbb{P}, \mathbb{P}'$ under $P, P'$ is as follows.
> > > Letting an event $\mathcal{E} = [J(T)=2, r_1 = O(\epsilon^2) ]$ (which occurs with $\mathbb{P}’[\mathcal{E}] = 1 - o(1)$ under instance $P'$),
> > > $\mathbb{P}[\mathcal{E}]
> > > = \exp\(-\sum_i N_i D(P_i'||P_i) )\mathbb{P}’[\mathcal{E}]\
> > > = \exp(- N_1 D(P_1||P_1’))\mathbb{P}’[\mathcal{E}]\
> > > = \exp(- O(\epsilon^2) T ) (1 - o(1))$.
> > >
> > >
> > > > Please explain the argument in Appendix D, especially, the expression after Eq. (9).
> > >
> > > Let $\tau$ be the stopping time of an algorithm for FC. This algorithm does a uniform exploration to build an empirical estimate $\hat{P}$. Equation after (9) states that the case the empirical estimate $[||\hat{P} - P || \ge c]$ is negligible for an arbitrarily small c>0. This is because
> > > $\mathbb{E}[\tau] = \mathbb{E}[\tau\ |\ ||\hat{P} - P|| < c ]\mathbb{P}[ ||\hat{P} - P|| < c ]
> > >               + \mathbb{E}[\tau\ |\ ||\hat{P} - P|| > c ]\mathbb{P}[||\hat{P} - P|| > c ] $
> > > For a sufficiently small $\delta$, $\mathbb{P}[ ||\hat{P} - P|| > c ] = o(1)$, and if $\mathbb{E}[\tau\ |\ ||\hat{P} - P|| > c ]$ is bounded as $O(\log(1/\delta))$, then
> > > $\mathbb{E}[\tau\ |\ ||\hat{P} - P|| > c ]\mathbb{P}[ ||\hat{P} - P|| > c ] = O(\log(1/\delta)) \times o(1)$
> > > which is $o(\log(1/\delta))$ and is small compared with the leading factor of $O(\log(1/\delta))$.

---

> > > > ### Comment · Reviewer_BbC9 · 2022-08-04
> > > > **Paper needs clarification**
> > > >
> > > > Even in the reviews, it seems there are many vague and hidden spots that need clarification. It seems a lot needs to be said to only motivate the example above. How can you improve the paper to explain the difference of FB and FC? I think the paper presentation needs extensive improvement.
> > > >
> > > > Examples of vague points:
> > > >
> > > > > In this allocation, it is easy to see that $r_2,r_3=\Theta(1)$ ...
> > > >
> > > > This sentence does not parse.
> > > > > Equation after (9) states that the case the empirical estimate [..] is negligible for an arbitrarily small c>0 ..

---

> > > > > ### Author Response · Authors · 2022-08-04
> > > > > **Additional clarification**
> > > > >
> > > > > > Examples of vague points:
> > > > >
> > > > > Derivation of $r_2, r_3=\Theta(1)$ may be indeed not straightforward, but we meant that we can confirm it by solving
> > > > > the optimization problem in Appendix D following the discussion in Garivier & Kaufmann (2016). We
> > > > > believe that giving the details of the computation for FC is out of the scope of this paper.
> > > > >
> > > > >  >This sentence does not parse
> > > > >
> > > > > Sorry for the typo. We meant that "the case that the empirical estimate becomes ... is negligible".
> > > > >
> > > > > Additional information missing in the last reply is as follows.
> > > > >  >  Kaufmann et al. (2016) consider the FB setting and derive a lower bound for it, which parse
> > > > > closely to your bounds.
> > > > >
> > > > > In Kaufmann et al. (2016) "On the Complexity of Best-Arm Identification in Multi-Armed Bandit
> > > > > Models", the results for FB setting with $K\ge3$ arms (Theorems 16 and 17, and bounds in a similar form
> > > > > also appear in, e.g., Audibert et al. (2010)) are not tight as discussed there. The main difference
> > > > > of our result is that the tight and achievable bound becomes available when we not only compare two
> > > > > instances $P$ and $P'$ but also compare all instances in $\mathcal{P}$, though the former comparison is
> > > > > sufficient to depict the failure of the optimal FC allocation.

---

> > > > > > ### Comment · Reviewer_BbC9 · 2022-08-07
> > > > > > **The detail of the FC example are very helpful**
> > > > > >
> > > > > > > We believe that giving the details of the computation for FC is out of the scope of this paper.
> > > > > >
> > > > > > As I mentioned earlier, I believe the paper is motivated by the fundamental difference between the FB and FC setting and a properly explained example like in Appendix D is required.

---

> > > > > > > ### Author Response · Authors · 2022-08-08
> > > > > > > **Re: The detail of the FC example are very helpful**
> > > > > > >
> > > > > > > Yes, the difference between FB and FC settings is the important motivation of our paper, and we will clarify it more in detail in the revised version. Still, the detailed *computation* appearing in the FC bound would not be so relevant and we will adequately skip it with a proper reference.

---

### Official Review · Reviewer_f8dM · 2022-07-10

**Rating:** 6
**Confidence:** 3
**Soundness:** 3 good
**Presentation:** 2 fair
**Contribution:** 3 good

**Summary:**

It is known that the probability of error in identifying a best arm in a fixed-budget $T$ is bounded by $\exp(-RT/H)$ where $H$ is a complexity term and $R$ quantifies the rate of decay as $T \rightarrow \infty$. This paper describes methods to possibly attain the best possible rate for the hardest instance in a given class, i.e., minimax optimal rate, which is called $R^{go}$. First a lower bound $R^{go}_B \leq R^{go}$ is described, next an algorithm with error rate $O(R^{go}_B - \epsilon)$ is described. Nevertheless, this algorithm assumes an $\epsilon$-optimal solution to an intractable sup-inf optimization and thus cannot be implemented. Consequently, a heuristic $R^{go}$-tracking algorithm using neural network (TNN) is described and compared to other methods in experiments.

**Questions:**

* Algorithm 2 describes one of the key contributions by introducing the "stored" queue of means but I'm unable to fully understand. In Algorithm 2 pseudocode the input is assumed to be an oracle that can solve for $\mathbf{r}^{B,*}$. From eq (3) the input to this optimization is the $\mathbf{P}$ and $\mathbf{Q}^B$ that attain the inf.

But in line 5 the allocation $\mathbf{r}^*_{b-K}$ is computed. How this step can be realized without knowing all the elements of $\mathbf{Q}^B$ which include $(\mathbf{Q}'_{b-K+1}, .. , \mathbf{Q}'_B)$ is unclear.

* It seems when $B=1$ the quantity $R^{go} = R^{go}_B$ and for $B > 1, R^{go}_B < R^{go}$. Then why are we interested in $R^{go}_\infty$, in the sense that it is going farther from the true oracle rate? Possible related, why is eq (21) true?

* How sensitive is the choice of $N^{true}$ and $N^{emp}$ for the performance of TNN algorithm?

**Limitations:**

None noted

**Strengths And Weaknesses:**

Obtaining good bounds on the optimal minimax rate has not been considered a lot in the existing literature, possibly because it is a hard resource allocation problem. Viewed in this light, the problem is similar to dynamic programming. However the rate $R$ is defined when the budget $T \rightarrow \infty$, and this brings it closer to prior work in best arm identification. Overall, the task considered in this paper is novel.

The claims made in the paper seem to be technically well-supported. However the same cannot be said about the remarks. For eg, remark 1 says $R^{go}$ may be 0. But shouldn't the choice of $\mathbf{Q}$ ensure at least one of the divergence terms is positive, and thus the outer sup over $\mathbf{r}$ would make the objective positive? Remark 2 mentions that a $P$ may be in $\mathcal{P}\setminus \mathcal{Q}$ without discussing the possible zero denominator in the KL divergence. Remark 3 mentions no algorithm can be better than DOT but I think the statement is only true for algorithms that are restricted to operate in a batched manner with equal-sized batches.

These comments are to the best of my understanding and I may have missed something as I felt the paper was generally quite hard to read. I do not understand the description in the introduction and section 1.1 about fixed budget BAI being impossible in an instance-wise optimal manner. For eg, the probability of error of successive rejects algorithm is at most $\exp(-T/(\log(K)H))$ which is optimal in order as described by Carpentier and Locatelli (2016). At some places, notation could be recapped to aid readability. For eg, in Theorem 2, it could be recapped that the domain of each $\mathbf{r}_i$ is all possible collections $(\mathbf{Q}_1, ..., \mathbf{Q}_i)$ to emphasize the batched nature of algorithms to which the lower bound applies. In the proof of Lemma 10 it will be better to describe the proof for the base case. The proof of Theorem 2 could be made easier to read by adding more explanations for the notation. It is not clear to me what $\mathbf{r}_1(\mathbf{Q}^1; \pi, \epsilon)$ means.

While the paper tries to address a hard problem, I believe some of the arguments need to be made more readable and explained bette for it to have a much higher impact.

---

> ### Author Response · Authors · 2022-08-01
> **Response to Official Review of Paper7640 by Reviewer f8dM**
>
> We thank the reviewer for the careful reading. We answer each question in what follows.
>
> > But in line 5 the allocation ${\mathbf{r}}^*_{b-K}$ is computed. How this step can be realized without knowing all the elements of $\mathbf{Q}^B$ which include $\mathbf{Q}_{b-K+1}',...,\mathbf{Q}_B'$ is unclear.
>
> The function $\mathbf{r}^*_b(\cdot)$ itself is predetermined before the first round. The allocation is determined by plugging in these arguments.
>
> > It seems when $B=1$ the quantity  $R^{go}=R_B^{go}$ and for $B>1$ ,  $R^{go}<R_B^{go}$. Then why are we interested in $R_\infty^{go}$ in the sense that it is going farther from the true oracle rate? Possible related, why is eq (21) true?
>
> First of all, we clarify that it remains unknown whether $R^{go}$ is achievable or not by a specific algorithm even if we admit infinite power of computation. Meanwhile,  $R^{go}_{\infty} (\le R^{go})$ is achievable by DOT and is the minimax optimal rate given $H(\mathbf{P})$.
>
> > How sensitive is the choice of $N^{true}$ and $N^{emp}$ for the performance of TNN algorithm?
> - Doubling or halving f $N^{true}$ and $N^{emp}$ did not change the performance so much. We will add some discussion on these parameters in the revised version.
>
>
> > Remark 1 says $R^{go}$ may be $0$. But shouldn't the choice of $\mathbf{Q}$ ensure at least one of the divergence terms is positive, and thus the outer sup over $\mathbf{r}$ would make the objective positive?
>
> For example, if we take $H(\mathbf{P}) = 1/\Delta$ (rather than $\Delta^2$ of $H_1$), in such a case rate $\exp(-R^{go}T/H)$ is not achievable for fixed $R^{go} >0$ around a small gap and $R^{go} = 0$ since the infimum over $\mathbf{P}$ is taken even if divergence becomes positive for each $\mathbf{Q}$.
>
> > Remark 2 mentions that a $P$ may be in $\mathcal{P}\setminus \mathcal{Q}$ without discussing the possible zero denominator in the KL divergence.
>
> The discussion in this paper covers the case of $D(\mathbf{Q}||\mathbf{P})=\infty$. This sometimes happens even if $\mathcal{P}=\mathcal{Q}$ when the probability measure $\mathbf{Q}$ is not absolutely continuous with respect to the measure $\mathbf{P}$.
>
> > Remark 3 mentions no algorithm can be better than DOT but I think the statement is only true for algorithms that are restricted to operate in a batched manner with equal-sized batches.
>
> This statement also covers the class of algorithms that are not limited in a batched manner, as shown in Theorem 2 (Proof is given in Appendix F.1), which uses the technique of change-of-measure for general algorithms.
>
> > I do not understand the description in the introduction and section 1.1 about fixed budget BAI being impossible in an instance-wise optimal manner. For eg, the probability of error of successive rejects algorithm is at most $\exp⁡(−T/(\log⁡(K)H))$, which is optimal in order as described by Carpentier and Locatelli (2016).
>
> The result of Carpentier+ means that no algorithm can achieve the error probability $\exp(-CT/H(\mathbf{P}))$ **jointly** for all instance $\mathbf{P}$, but it allows an algorithm with an error probability better than $\exp(-CT/H(\mathbf{P}))$ for some $\mathbf{P}$. Here, how an algorithm can perform better for some $\mathbf{P}$ depends on the performance required for another instance $\mathbf{P}'$ (in other words, there is a trade-off between performances on $\mathbf{P}$ and $\mathbf{P}'$). This is why we need $H(\mathbf{P})$ to describe an optimal algorithm, and in this sense, the instance-wise optimality cannot be expected for the FB setting.
>
> > The proof of Theorem 2 could be made easier to read by adding more explanations for the notation.
>
> Thank you for the careful reading. We will add additional clarification on the proof of Theorems 1 and 2.

---

> > ### Comment · Reviewer_f8dM · 2022-08-07
> > **After reading author response**
> >
> > Thank you for the clarifications. With regards to $B$, I think $R^{go} = R^{go}_1$ comparing eq (2) and eq (3). However I believe authors misquoted me in the response, I had said $R^{go}_B \leq R^{go}$
> >
> > If that is correct, is the reason why we are interested in $R^{go}_{\infty}$ is because it is the best we can do on the worst problem instance?

---

> > > ### Author Response · Authors · 2022-08-08
> > > **Thank you for the confirmation**
> > >
> > > > I had said $R_B^{go} \le R^{go}$
> > >
> > > Sorry for the misquotation in our author response. You are right, our Corollary 3 states that $R_B^{go} \le R^{go}$.
> > >
> > > > If that is correct, is the reason why we are interested in $R_\infty^{go}$ is because it is the best we can do on the worst problem instance?
> > >
> > > Yes, $R_\infty^{go}$ is the best rate against the worst problem instance (minimax rate).

---

### Official Review · Reviewer_RNv6 · 2022-07-13

**Rating:** 7
**Confidence:** 4
**Soundness:** 3 good
**Presentation:** 4 excellent
**Contribution:** 3 good

**Summary:**

The authors consider the problem of best arm identification in the fixed budget setting. The authors introduce a notion of global optimality specific to a prescribed complexity measure $H$. They establish a characterization of the optimal rate fo such notion via the rate $R^{go}$ (see Theorem 1) and provide Algorithm 1, which samples by tracking an $\epsilon$ solution of the optimal allocation that achieves $R^{go}$. While algorithm 1 remains infeasible since it requires accesss to unavailable information, the authors propose weakened characterizations of the optimal rate $R^{go}$ via the rates $R^{go}_B, R^{go}_\infty$ (See Theorem 2). These new rates take into account the fact that an algorithm throughout its learning may not have information about future empirical mean estimates of the arms. Then the authors provide Algorithm 2 (Delayed Optimal Tracking) which enjoys a guarantee that matches the lower bound of Theorem 2 (see Theorem 5 and Theorem 7). The proposed algorithms so far do require solving a nontrivial optimization problem. To remedy this, the authors propose an approach that uses a neural network to learning the optimal solution of such optimization problems, and provide along with that numerical experiments to consolidate their approach.


**Questions:**


- How could one choose the complexity measure? The authors claim that one cannot obtain an instance specific lower bound as in fixed confidence setting, which I am not convinced of. However, if this is the case, does this suggest some sort of trade-off when choosing what complexity measure to focus on? Could the authors comment on this?
- By now we do have practical algorithm for the fixed confidence setting. Is there a big gain from tracking an allocation determined from solving the fixed budget lower bound instead of the fixed confidence lower bound? In Kaufmann et al. it has been shown that in the two armed bandit for the class of gaussian distributions the complexity measures that arise from both problems are the same, so perhaps there is no gain, and one could simply track allocations from the fixed confidence setting. Could the authors comment on this?
- As far as I understand the results are only valid when $\mathcal{Q}$ is restricted to the class of Gaussian distributions or Bernoulli distributions but not both at the same time. Could the author comment on this?


**Limitations:**

I think the authors have properly addressed the limitations of their work both theoretically and practically. The work remains theoretical at large, so I don't see any potential negative societal impact or ethical issue that apply.

**Strengths And Weaknesses:**

Strengths


- This paper is well written paper with solid theoretical results, good literature review, and a clear presentation. This makes the paper easy to follow even though it is quite technical.
- The problem considered by this work lacks considerable understanding in the literature and as highlighted by the authors remains poorly understood. Therefore, the theoretical results of this paper, especially the characterisations of the lower bound in section 2 (Theorem 1 and 2) appear to be worthwhile contributions in my opinion that further our understanding in the fixed budget setting.
- It is really appreciated that that the authors did not stop at providing a perhaps theoretically good, but computationally infeasible algorithm. Instead the authors proposed a neural networked based learner to remedy this challenged and accompanied that with numerical experiments.


Weaknesses

- I am not fully convinced about the discussion in section 1.1. (lines 42-51) that one has to choose a complexity $H(\cdot)$ to characterize optimality in the fixed budget setting. I do not have an answer but at the same time this opens the question of how one should choose the the complexity measure $H(\cdot)$ which I think the author could clarify further. Is there a better complexity measure? are they even comparable in some sense?
-  Another weakness of the current work is that the proposed algorithms whether they rely on neural networks or not remain largely impractical (the authors also highlighted this).

---

> ### Author Response · Authors · 2022-08-01
> **Response to Official Review of Paper7640 by Reviewer RNv6**
>
> We thank the reviewer for many insightful comments. We answer the questions in what follows.
>
> > How could one choose the complexity measure? … However, if this is the case, does this suggest some sort of trade-off when choosing what complexity measure to focus on?
>
> There is indeed a kind of trade-off. In other words, the "optimal" algorithm (and therefore its performance for some specific instance) depends on the choice of $H(P)$. Maybe a natural choice of $H(P)$ would be as follows. When using the optimal algorithm DOT for some $H(P)$, the exponent of the error probability is $R_{\infty}^{go} / H(P)$ for the worst instance $P$. We think that the natural choice of $H(P)$ would be such that the exponent of DOT becomes $R_{\infty}^{go} / H(P)$ for **all** $P$ since $H(P)$ should capture all information on the difficulty, but the existence of such $H(P)$ is extremely nontrivial and would go beyond the scope of this paper.
>
> > By now we do have practical algorithm for the fixed confidence setting. Is there a big gain from tracking an allocation determined from solving the fixed budget lower bound instead of the fixed confidence lower bound?
>
> The fixed-budget BAI and fixed-confidence BAI become essentially different for $K\ge3$. Some events that are negligible in the asymptotics on the confidence level matter in the fixed-budget setting, and it largely affects the algorithm design. We show a $K=3$ instance below where the fixed-confidence optimal algorithm performs poorly. Let $P = (0.6, 0.5, 0.5-\epsilon)$ and $P' = (0.4, 0.5, 0.5-\epsilon)$. Let $A(\delta)$ be a parametrized algorithm. Assume that we run $T$ rounds the parameter $P'$ and this algorithm draws in accordance with the optimal rate of the fixed confidence BAI (GK 2016, (8) in our supplementary). In this case, it would only draw arm 1 for $O(\epsilon^2)$ times and recommend arm 2 with probability $1-o(1)$. A standard change of measure argument states that if we run $A(\delta)$ on the parameter $P$, with a probability at least $(1-o(1))\exp(-O(\epsilon^2)T)$ it recommends arm 2 (which is incorrect under $P$), and thus the rate is only $\exp(-O(\epsilon^2)T/H_1(P))$, which can be arbitrarily worse by taking small $\epsilon$ in view of the fixed-budget BAI.
>
> > As far as I understand the results are only valid when $\mathcal{Q}$ is restricted to the class of Gaussian distributions or Bernoulli distributions but not both at the same time. Could the author comment on this?
>
> With probability 1, we are able to identify whether the distribution is Bernoulli or Gaussian, and thus our algorithm can deal with both at the same time by using the same discussion. The specific value of $R^{go}$ changes depending on the considered model.
>
> > Another weakness of the current work is that the proposed algorithms whether they rely on neural networks or not remain largely impractical (the authors also highlighted this).
>
> We agree that scaling up the neural network algorithm remains a challenge. This is due to the hardness of the complicated minimax optimization problem in the theoretical bound.

---

> > ### Comment · Reviewer_RNv6 · 2022-08-03
> > **Further questions and comments**
> >
> > **About the example between FB and FC.** When you say "... certain events are negligible in the asymptotics ..." can you precise what events? I presume you mean the events of recommending a bad arm.  Could you also clarify asymptotic in what sense? is it as $T \to \infty$ or  $\delta \to 0$? I don't quite get the conclusion at the end of your example. You are saying that sampling under the FB allocation will give us a lower bound via a change of measure argument under P that is $(1-o(1))\exp(-O(\epsilon^2)T)$. This I agree with. Then, you suddenly mention the probability of error rate $\exp(- O(\epsilon^2)T/H_1(P))$ which would be the one under the fixed budget in your example. Is this correct?  How am I to compare the two. The dependence in $epsilon$ is the same. Are you claiming that there are regimes of  $H(P)$ that would make one better than the other? If so the opposite is also true.
> >
> > **About distinguishing Gaussian and Bernoulli.** I see this was kind of easy because the Bernoulli distribution is discrete. I was thinking more generally, can your results be extended beyond a specific class of distributions, say the exponential family which encompasses both gaussian and Bernoulli but also other distributions.

---

> > > ### Author Response · Authors · 2022-08-04
> > > **Answer to the further questions**
> > >
> > > Thank you for the additional questions. We answer to these questions in what follows.
> > >
> > > > When you say "... certain events are negligible in the asymptotics ..." can you precise what events? I presume you mean the events of recommending a bad arm.
> > >
> > > Following the FC notation (Section D), let $\hat{P}$ be the empirical estimate of the true parameter $P$. We intend event $\mathcal{E} = [||\hat{P} - P|| > c]$ for an arbitrarily small constant $c>0$. Such an event does not affect the achievable FC rate. In particular, letting $\tau$ be the stopping time of an FC algorithm,
> > > $\mathbb{E}[\tau] = \mathbb{E}[\tau\ |\ ||\hat{P} - P|| < c ]\mathbb{P}[ ||\hat{P} - P|| < c ] + \mathbb{E}[\tau\ |\ ||\hat{P} - P|| > c ]\mathbb{P}[ ||\hat{P} - P|| > c ]$.
> > >
> > > Then the second term of RHS is
> > > $\mathbb{E}[\tau\ |\ ||\hat{P} - P|| > c ] \times \mathbb{P}[ ||\hat{P} - P|| > c ] = O(\log(1/\delta)) \times o(1) = o(\log(1/\delta)), $
> > > which is negligible compared with the first (leading) term of RHS.
> > >
> > > > Could you also clarify asymptotic in what sense? is it as $T \rightarrow \infty$  or $\delta \rightarrow 0$?
> > >
> > > Asymptotic means that the limit of $\text{(required number of samples)}/\log(1/\delta)$ as $\delta \to 0$ in FC and that of $\text{($\log$ PoE)}/T$ as $T \to \infty$. In FB, $\mathbb{P}[\mathcal{E}]$ occurs with probability $\exp(-O(T))$, and thus we need to consider this event to describe the minimax rate.
> > >
> > >
> > > > I don't quite get the conclusion at the end of your example. You are saying that sampling under the FB allocation will give us a lower bound via a change of measure argument under $P$ that is $(1−o(1))exp⁡(−O(\epsilon^2)T)$. This I agree with. Then, you suddenly mention the probability of error rate $\exp⁡(−O(\epsilon^2)T/H_1(P))$ which would be the one under the fixed budget in your example. Is this correct? How am I to compare the two. The dependence in $\epsilon$ is the same. Are you claiming that there are regimes of $H(P)$  that would make one better than the other? If so the opposite is also true.
> > >
> > > As you suggested, the dependence of $(1−o(1))\exp⁡(−O(\epsilon^2)T)$ and $\exp⁡(−O(\epsilon^2)T/H_1(P))$ on $\epsilon$ is the same because $H_1(P) = 1/(0.6-0.5)^2 + 1/(0.6-0.5-\epsilon)^2 \approx 200 = O(1)$. We meant that this error probability of form $\exp(-O(\epsilon^2)T/H_1(P))$ is arbitrarily worse than the FB allocation (like successive rejection and DOT) with error probability $\exp(-O(1) T/H_1(P))$.
> > >
> > > > About distinguishing Gaussian and Bernoulli. I see this was kind of easy because the Bernoulli distribution is discrete. I was thinking more generally, can your results be extended beyond a specific class of distributions, say the exponential family which encompasses both gaussian and Bernoulli but also other distributions.
> > >
> > > Theorem 1 and 2 can be generalized by setting $\mathcal{P}$ as the set of all possible distributions. We believe DOT (Algorithm 2) can be generalized as well, but it is limited to Gaussian with a fixed variance and Bernoulli so far.

---

> > > > ### Comment · Reviewer_RNv6 · 2022-08-05
> > > > **Comment**
> > > >
> > > > Ok, I see. Thanks for the clarification. I may come back to you with more questions upon further inspection of the manuscript and the other reviewers comments.
> > > >
> > > > It is a bit strange that your events are based on the error between the probability distributions. It is not difficult to see that there is a set of distributions P that have the same best arm. Therefore, as long as you have an estimate that is within such set, the estimation error shouldn't matter the much any more, assuming the set is compact and say convex. I will reflect upon this further.

---

> > > > > ### Author Response · Authors · 2022-08-07
> > > > > **Thank you for comment**
> > > > >
> > > > > Thank you for the comment. As pointed out, there are many candidates of distributions $P$ that have the same best arm as the true distribution. It is true that the estimation error (within such a set) should not matter when we determine the recommendation arm after $T$ rounds. On the other hand, our results suggest that we should deliberately determine arms to explore reflecting even a slight change in the estimated distributions.

---

### Meta-Review · Area_Chair_c4L8 · 2022-08-20

**Recommendation:** Accept
**Confidence:** Less certain

**Metareview:**

This paper had very mixed pre-rebuttal scores, fairly detailed reviews, and significant author/reviewer interaction.  Following all of this, the reviewers are now generally positive, with several of the initial concerns being resolved.  One of the scores remains below the threshold, due to certain claims and statements being too vague, and insufficient distinction between fixed confidence and fixed budget.  However, another reviewer responded by noting that the distinction is generally clear from existing works (e.g., [a]).

Since this remaining concern does not appear to be a deal-breaker, I believe that acceptance is the correct decision.  However, the authors should very carefully modify the paper according to the reviewer feedback, and be extra careful of unclear statements such as those pointed out by Reviewer BbC9.

[a] Emilie Kaufmann, Olivier Cappé, and Aurélien Garivier. On the complexity of best-arm identification in multi-armed bandit models. JMLR, 2016.

**Award:**

No

---

### Decision · Program_Chairs · 2022-09-14

Accept